# Rush order containment of critical drugs in ICUs

**Paola Cappanera**[1]*, **Maddalena Nonato**[2], **Filippo Visintin**[3], **Roberta Rossi**[1]

**1** Dipartimento di Ingegneria dell'Informazione, University of Florence, Florence, Italy, **2** Dipartimento di Ingegneria, University of Ferrara, Ferrara, Italy, **3** Dipartimento di Ingegneria Industriale, University of Florence, Florence, Italy

* paola.cappanera@unifi.it

**Data Availability Statement:** Data are available from https://github.com/pcfvmn/PLOSONE.

**Funding:** The work by P. Cappanera, R. Rossi and F. Visintin has been partially supported by two

## Abstract

The recent SARS CoV-02 pandemic has put enormous pressure on intensive care staff, making it imperative to relieve them of repetitive tasks with little added value such as drug replenishment. We propose a decision support system based on a hybrid policy to manage the inventory of critical drugs with low and intermittent demand at an Intensive Care Unit (ICU). Demand forecasting is at the heart of any inventory policy. We claim that in the ICU setting drug demand patterns must be therapy based. Heterogeneous data have been collected during an on site study, and information have been extracted to provide a faithful abstract representation of the ward as a system, as well as the potential evolutions of ICU patients clinical conditions. Together with medical guidelines, this provides the foundation of a therapy based demand forecasting tool. This study integrates schedule optimization and demand forecasting, and exploits simulation for evaluation purpose in the long run. At the beginning of every period, drug orders are optimally scheduled with respect to forecast demand. Then, scheduled orders are deployed day by day and confronted with the real demand in a simulated environment. Potential stock outs trigger rush orders to restore safety stocks. The comparison between the proposed policy and a standard policy mimicking current practice in an ICU ward shows that information on therapy patterns can be successfully incorporated into drug replenishment processes to reduce the number of rush orders, a primary goal in designing an efficient system.

## 1 Introduction

In most developed western countries, costs related to drug logistics make up an ever growing share of hospital expenses. According to [1], retail pharmaceutical spending has grown slowly or even declined across Europe in the last decades; nevertheless, hospital pharmaceutical spending has steadily expanded, raising a financial sustainability issue regarding public health. Saving opportunities rely on improving the different components of the drug supply chain and on coordinating their integration [2]. Inventory management at point of care, in particular, can be a potential source of inefficiency. This is not only due to frequent overstocking or stock out-driven emergency orders, but it is also related to the time highly trained personnel

projects: (i) the project Optimized management of drug orders in ICUs, funded by Ente Cassa di Risparmio di Firenze, 2014 (https://www.fondazionecrfirenze.it/bandi-e-contributi/), and (ii) the LINFA (Logistica INtelligente del FArmaco) project, funded by Regione Toscana under the call PAR FAS 2007-2013, Linea d'azione 1.1 - Bando FAR FAS 2014 (https://www.regione.toscana.it/-/bando-far-fasfinanziamento-di-progetti-di-ricerca-nei-settori-energia-fotonica-ict-e-robotica). The funders had no role in study design, data collection and analysis, decision to publish, or preparation of the manuscript.

**Competing interests:** The authors have declared that no competing interests exist.

spend looking after frequent replenishment operations. The recent SARS CoV-02 pandemic has put enormous pressure on intensive care staff, making it imperative to relieve them of simple repetitive tasks with little added value such as drug replenishment. Moreover, the waste commonly associated with poor drug management in ICU is even more socially and economically unacceptable in light of the dramatic situation we are experiencing.

In light of the above mentioned considerations, decision makers in health care organizations have started looking towards inventory theory in search for well-assessed practices to hedge against the rise of drug inventory costs. Most of these practices come from the manufacturing industry. Drug inventory management, indeed, shares some of the most challenging features of inventory problems arising in manufacturing, such as (*i*) limited storage areas, (*ii*) multi-item inventory, (*iii*) intermittent demand, (*iv*) different sizes of units used in consumption (dosage) with respect to replenishment (boxes), and these data may even vary from drug to drug, as well as (*v*) the need to take into account different stakeholders in the decision process. Despite such common issues, though, drug inventory presents peculiarities that make it difficult to transfer to the health care industry the best practices that have proven successful in manufacturing. In particular, (*vi*) since patient health is the primary goal, service level prevails over revenue in performance evaluation; (*vii*) since back orders are not allowed, stock out events trigger expensive rush orders.

This work concerns drug replenishment policies at an Intensive Care Unit (ICU), with a focus on antibiotics. In ICU settings, the above mentioned features are exacerbated since: most patients are in critical conditions, drug shortages may have dramatic consequences on fragile patients, and staff work under tight time constraints. Regarding the search for consensus (*v*), in particular, ICUs are sensitive realities in which the involvement of different stakeholders with possible conflicting priorities plays a pivotal role; hospital management, clinicians, and nurses have to coordinate complex and strictly intertwined decisions while pursuing a shared patient-centered perspective. Drug inventory management is no exception, as each stakeholder evaluates an inventory policy according to his/her own potentially diverging perspective [3]. Two features that usually characterize the ICU wards complicate matters further: the limited number of hospitalized patients and the frequent changes in therapy they are subject to. Both these features cause high demand variability over time, with different demand patterns for each drug. Such high demand dynamicity, on the one side encourages to keep hidden stocks at ward as a means to promptly react to a therapy switch, on the other side it makes drug inventory management at ICUs a labour intensive activity that erodes nurse time to the detriment of patient care.

In our case study, the process is ruled according to the following procedure. ICU's regular orders for drugs are forwarded to the hospital pharmacy and are fulfilled within a short lead time (one or two days). In case of stock out, though, emergency (rush) orders are forwarded to the upper tier of the distribution chain (a regional warehouse) and are fulfilled within shorter lead times (typically within few hours) at a much higher cost. This cost has a substantial impact on hospital expenditure and its containment plays a crucial role in the design of efficient replenishment policies.

In an effort to reduce drug-related costs, this work focuses on capping rush orders. To this aim, the role of demand forecast is crucial. The direct observation of the processes related to drug management at the ICU of a large public hospital in Italy suggests that drug consumption is therapy driven rather than drug driven. More precisely, the total quantity of drugs consumed in one day is of scarce utility to predict drug consumption in the following ones. The drug demand on a certain day, instead, depends on the therapies prescribed to the hospitalized patients in the previous days. We conjecture that stock outs may be reduced while still keeping the inventory level below a reasonable threshold, thanks to the exploitation of available

knowledge on the therapies that drive drug consumption. To the best of our knowledge, we are not aware of studies addressing drug replenishment in ICUs while leveraging a therapy-driven demand. On the other hand, many studies [4–6] claim that it would be extremely useful for ICUs to have tools to predict the evolution of patients' clinical conditions. We believe that predicting demand is closely linked to predicting the evolution of the patient's condition which motivates our study. As a byproduct, this study provides the building block of a decision support system due to assist nurses in charge of drug logistics.

We discuss an integrated tool consisting of the following three components: *the predictor*, *the optimizer*, and the *simulator*.

The predictor is built upon a formalization of the decision process driving the demand of antibiotics and it is used to forecast the demand of drugs in a given time period, hereinafter referred to as the *planning period*. For each period, prediction relies on the status of the current patients. Specifically, the predictor assumes that patients are clustered in groups according to the severity of their clinical conditions at admission, and for each patient cluster it generates a set of paths representing the possible evolutions of their clinical conditions. Paths are generated by combining together two kinds of information: (*i*) medical guidelines that suggest care plans, and (*ii*) patient-related information. The paths, which demand generation relies upon, revealed to be a crucial structure in capturing the peculiarity of the care process.

Once the predictor has issued a scenario, its demand forecast is forwarded to the optimizer which computes an optimal drug order agenda for the planning period, based on the current stock levels. The optimizer solves an inventory problem at point of use by means of a Mixed Integer Linear Programming (MILP) model inspired by the Lot-Sizing problem.

Based on empirical distributions, the simulator generates patients, assigns each generated patient to a cluster and assigns her/him a therapy. In addition, it reproduces the drug re-stocking process. Orders may either anticipate demand (what in manufacturing is known as push orders) or fix a shortage (the equivalent to pull orders in manufacturing). The former follow the drug order agenda defined by the optimizer. The latter, instead, are issued to restore safety stocks and to deal with stock outs. The larger the discrepancy between forecast and actual demand, the larger the number of pull orders that will be issued.

Each simulation covers a *scheduling period* which is not longer than the planning period. At the end of the current scheduling period the whole mechanism is iterated, providing the forecast tool with the updated data about patients and inventory. Given a planning period, the closer the length of the scheduling period to the length of the planning one, the greater the confidence in the prediction tool.

This hybrid push-pull policy, which aims at order reduction and regularity, is tested against a variant in which the forecast is not therapy driven but drug driven. In the competitor, the forecast tool relies on a Monte Carlo process whose parameters have been empirically estimated (based on the average demand for each drug). With respect to our forecasting method, the Monte Carlo-based predictor considers, for each drug and for each day, the probability that the number of doses administered will be equal to a discrete number. Such a predictor does not consider neither patients' therapies nor the drugs administered on previous days. Finally, both variants, i.e., the integrated tool based on the therapy-driven predictor and the competitor tool based on a Monte Carlo process, are tested in the long run against a fixed review period strategy that is widely used as common practice. The latter consists of placing regular orders once a week, according to an ($S$, $s$) pattern, i.e., if stock level is below $s$ an order up to $S$ is issued, and resorting to extra orders when required. The computational results support our claim about the importance of incorporating therapy-based demand forecasting in the designed policy.

In summary, the contributions of this work are as follows: The study (*i*) integrates scheduling tools and demand forecast in a rolling horizon framework, (*ii*) provides a prediction tool that reflects the peculiarities of drug consumption whenever the process is therapy driven rather than drug driven, (*iii*) gives evidence of the relevance of considering therapies instead of independent drugs in reducing the number of stock outs and consequently costs, (*iv*) provides a comparison between two integrated tools based on different prediction components, (*v*) tests the two resulting integrated tools against a classical fixed review period policy that mimics the current practice in wards, and (*vi*) is based on real/realistic data collected at a large public hospital in Italy and on a deep knowledge of the complex processes regulating the ward.

The rest of the paper is organized as follows: related works are reviewed in Section 2, with a focus on those devoted to inventory policies at point of use in health care. Our case study is introduced in Section 3. Section 4 is devoted to the demand forecasting process. The proposed solution approach is presented in Section 5 and results are discussed in Section 6. Section 7 briefly describes a preliminary decision support system, and, finally, future research directions are discussed in Section 8.

## 2 Literature overview

Research in the area of health care inventory has been thriving, as testified by an ever increasing body of literature published in the last two decades. In this setting, cost containment is of paramount importance since expenditure on medical supplies, equipment, and pharmaceuticals is second only to personnel costs [7]. However, in the health care setting the supply chain management has a more recent history than in other industries [8]. Consequently, many authors see leveraging cutting edge supply chain management practices as an effective strategy to realize cost containment in hospital logistics and procurement. Some recent reviews provide an updated selection of the most significant contributions. In particular, [2] surveys state-of-the-art research on material logistics in hospital, including pharmaceuticals, covering the period from 1998 to 2014; more recently, [9] addresses medical supply logistics with a focus on the operating room environment; [10] underlines the criticalities of pharmaceuticals inventory in case of personalized treatment plans both on patients and on service providers, whereas [11] broadens the analysis to encompass the distribution of medical supplies from the central pharmacy to the point of care. The reader is addressed to these papers for a broad picture of health care logistics. Hereafter, we focus on studies presenting pharmaceutical inventory policies at point of use and highlight, when present, the integration of demand forecasting tools in the inventory policy.

Inventory policies are influenced by the technology implemented to keep track of inventory, by the process automation level and by the integration of the information systems involved, if present. In the recent past, nurses used to be in charge of all inventory and replenishment duties often without the support of information technology [12]: they would conduct periodic eyeballing surveys of inventory, anticipate future needs based on common practice, and issue replenishment orders on the basis of such empirical demand forecasting. This role, which is extremely demanding and time consuming, has been progressively replaced by the adoption of so called *cyclic policies*. These are based on a combination of review frequency and ordered quantity. The most common one is the so called *order up to* policy ($T$, $S$), a periodic review in which every $T$ periods what is needed to restock up to a level $S$ is ordered. Cyclic policy adoption moved the replenishment decision-making task away from clinical personnel to a centralized administrative body, in charge of sizing the best parameters [7]. The evolution of technology, such as the introduction of barcodes and radio frequency identification (RFID) systems [13], automatized the time consuming visual evaluation of the items in need for

replenishment and made perpetual inventory possible, i.e., real-time perfect knowledge of the current and incoming stock levels. This allowed policies based on continuous review, according to which the current stock level is compared to fixed quotas and orders are triggered accordingly: for example, in the $(s, S)$ policy as soon as inventory goes below $s$ an order up to level $S$ is issued, while in the $(s, Q)$ policy a fixed quantity $Q$ is ordered. Also the adoption of *Automated Point of Use* (APU) systems such as the *Automated Dispensing Cabinets* (ADCs) make it possible continuous review policies [14]; indeed, such a technology allows to store drugs and data concerning the prescribed therapies, to automatically update the inventory level and place orders whenever the stock level of an item drops below its $s$ threshold. However, ADCs are not fit for the ICUs setting as ICU's demand is dynamic and intermittent. On the contrary, ADCs are suitable in case of regular demand, high consumption rate, and high level of automation (see [15] for a discussion on the Artima system and [16] for a discussion on the use of the Pyxis Med Stations).

Many papers study cyclic inventory policies under the following assumptions: (*i*) the time interval between demand events of the same drug is exponentially distributed, (*ii*) daily consumptions of different drugs are independent, and (*iii*) demand is stationary over time. The abovementioned review [10] searches the literature on pharmaceutical inventory in hospital pharmacies regarding how often periodic replenishment should be performed, which policy to adopt, and how to set the thresholds that govern decision making in these policies. As an example, [17] addresses the inventory of disposable items at point of use in case of tight storage capacity and short lead time, aiming to maximize service level and compares three periodic review policies.

Others explore the effects of introducing additional information such as drugs criticality and expiration date within this classical framework. For example, a comparative study of information-aware policies conducted in [7] shows that the most informed policy provides the highest cost containment but stock outs often arise. The authors advocate the need of demand forecasting tools incorporating information about the number of patients in the ward and the prescribed drugs, as a way to reduce stock outs. In [18], a hybrid policy is discussed, which combines periodic and continuous policies: ADMs are restocked periodically according to a $(s, S)$ policy, whereas whenever inventory falls below a critical level $R$, an urgent replenishment order of $Q = S - R$ is issued to avoid stock outs. The study focuses on a single item with stationary demand, and proposes a simulation-optimization-based heuristic to estimate the parameters' value. In a later work [19], the same authors explore the so called 2-bin policy, according to which replenishment orders are triggered proactively, based on inventory levels.

The development and integration of patient based demand forecasting plays a pivotal role to address the challenge of non stationary demand. This may be realized in many different ways.

In [20], demand forecasting of a chemotherapy drug is based on real-time availability and reliability of data concerning patients and therapies, and on the formalization of the rules that govern decision making in that clinical setting.

Reliable real time information is also envisioned in [21] where look-back and look-ahead policies are discussed. Integrated information flows are exploited to pull replenishment at points of use and at the hospital pharmacy. By discrete event simulation, a Periodic Automatic Replenishment (PAR) policy is compared with a look-ahead policy based on patients' needs as derived from medical prescriptions, for 19 drugs and 3 medical units. Drug demand is modeled as a Monte Carlo process based on the known demand frequency distribution and assuming the uniform usage of drugs throughout the consumption period. The look-ahead policy allows significant savings, provided real-time information flows are available.

In [22] the authors report their experience for a non stationary demand case. In case of a perishable drug, the number of patients currently on therapy have been incorporated into the state space of the Markov decision process that models the system.

Time series are exploited in [23] where drug demand forecasting is based on temporal pattern matching and in [24], where inventory forecasting for a medical store is based on data mining techniques applied to transactional data of medical consumptions. A data mining technique is also used in [25] to identify temporal relationships between drugs prescribed to diabetic patients.

As a final remark, [2] acknowledge the potential of demand forecasting based on clinical guidelines applied to current patients, together with information flow integration. Research on the implications of demand forecasting on inventory policies is advocated, with the aim to address the correlation among items and low turnovers.

When the perspective is expanded to encompass general inventory problems, many more attempts have been made to take advantage of any information over the process that drives demand, on the premise that demand forecast integration within an inventory policy is crucial in improving the efficiency of the resulting system. An option is to dynamically vary the value of the parameters used in cyclic policies, i.e., the reorder point $s$ and the order-up-to level $S$, as discussed in [26]. This suggests that more complex situations, such as those with interdependent and nonstationary demand, may benefit from other means of knowledge representation. Recently, expert systems have been proposed to extract useful information on the decision process that rules demand. In particular, [27] investigate non-probabilistic inventory control strategies and propose *a belief-rule-based inventory control method* which is initialized by expert knowledge and historical demand information and trained over several time periods, to yield the rules that determine the optimal order quantity given current inventory and short-term demand forecast.

In summary, from the literature concerned with pharmaceutical inventory at the point of use emerges that: (i) information-aware replenishment policies are still scant and most needed; (ii) focussing on the process that generates the data (drug consumption) rather than on data themselves represents a promising avenue for the development of these policies; (iii) efficient policies should rely on forecasting tools taking into consideration the patients currently hospitalized, the therapies they are subject to, as well as the complex decision process regulating the drug prescription and administration, and finally (iv) as far as critical settings are concerned, these policies should, first and foremost, aim at minimizing stock outs and, consequently, rush orders.

In this paper we present a novel hybrid policy that meets these requirements and we evaluate its effectiveness via simulation.

## 3 Problem statement

In this section, we describe the addressed problem in terms of objectives and constraints, according to our observation of the ICU of a large public hospital in Italy. The drug management issues characterizing this ICU, however, are common to most of the ICUs we are aware of and can be summarized as follows:

1. Rush orders tend to be frequent and their fulfilment is very expensive. In case of antibiotic stock out, patients suffer from delayed treatments, and while waiting for the required drugs, they need continuous attention from nurses. The fulfillment of a rush order, in general, causes coordination problems (and cost) all along the supply chain. In our case, rush orders are fulfilled within two hours by a regional warehouse located 15 kilometers away from the ICU.

2. A lack of effective traceability systems, coupled with the presence of multiple stocking points, often dislocated far away one another, implies that nurses spend much time in non-value-added activities, such as stock level checking, drug ordering and drug restocking.

3. There is a need to contain the operating working capital by reducing the amount of drugs kept in stock. However, the need to ensure the availability of life-saving drugs inevitably makes the cost containment issue less important than in other settings.

4. The unpredictability of the patients' response to certain drugs and the (rather common) presence of comorbidities often determine ongoing changes in the therapies being prescribed, inevitably resulting in a large variability in patients' Length of Stay (LoS) and in the drugs administered.

Consistent with this scenario, our first goal is to decrease the number of rush orders.

The second goal is to reduce the time nurses spend in non-value-added activities. This goal is pursued by grouping orders on the same day, by minimizing the number of days in which an order occurs, and by looking for regularity in the process. Note that by assuring the first goal we also address the second one as we reduce the burden of rush orders that is in charge to nurses.

Third, cost-control is achieved by setting a soft upper bound to the operating working capital which is calculated as the number of drug boxes in stock times their cost.

In our case study, we deliberately focus on a restricted set of antibiotics considered critical by medical staff because of (*i*) their price, which makes hidden stocks prohibitive, (*ii*) their low and intermittent demand (they are selectively used in specific therapies devoted to patients who show up irregularly), and (*iii*) the fact that these antibiotic therapies require a continuous and regular treatment to succeed (intermittent use of the same drug is highly discouraged).

Unfortunately, infrequently used items challenge well-assessed inventory strategies in manufacturing, such as Kanban and Lean Six Sigma, which work well for items that are regularly used. In addition, drug demand should not be analyzed individually, as the demand for each drug is governed by therapy plans, and each plan consists of several drugs; therapies, in turn, are triggered by patients' arrival in the ward and by their evolution during their hospitalization. For these reasons, we do not apply classical inventory policies whose parameter estimation (e.g., $s$, $S$, $R$, and $Q$) relies on stationary demand and independence [28].

In terms of constraints, we model a quite general situation that can occur when drugs have alternative storage spaces. Specifically, in our case study, each drug has its own dedicated storage space in a medicine cabinet located in the inpatient room for prompt use as well as a shared storage space in other shelves and cabinets. Specific storage requirements, such as refrigeration, are considered. In these premises, we assume that drugs are partitioned in groups, i.e., drugs in the same group share a dedicated storage unit. The pharmaceuticals considered here have long expiration windows, so we disregard the perishability issue contrarily to what done in [29, 30] in a different setting.

Also, we consider the general case in which lead time may vary from day to day; as an example, in our case study, the lead time is one day but on a Saturday, while drugs ordered on Saturday will be available on the next Monday. Contrarily, urgent orders can be triggered anytime. The details are provided in Section 5.1.

In addition, the literature advocates the importance of involving several stakeholders in complex decision process such as the one concerning drug inventory. As an example, in our case study, nurses, hospital management, and clinicians play a pivotal role. Here, we build on a previous work [3] in which we formalized their preferences with respect to scheduling policies

and we investigated their mutual interaction. Motivated by this previous study, we select the most influential stakeholder and include its perspective in our decision-making process.

In summary, our policy is driven in a hierarchical way by rush order reduction, minimization of order events, and cost containment. While the hierarchy among the criteria may vary from one setting to another, we believe that all of them are shared by public health care facilities. The same holds for the constraints considered.

For all these reasons, we propose an ad hoc solution approach for the inventory management of the critical drugs tailored to ward features, whose main components are detailed in the next sections.

## 4 The demand generation process: From data collection to model building

### 4.1 Data collection

Hospitals in general, and the ICU considered in our case study in particular, are usually characterized by several heterogeneous data sources. In our study, we consider the following 5 data sources:

1. Prosafe data [31]: prosafe is the critical care information system in place over the data collection period—digital data

2. data from the microbiology laboratory—digital data

3. data on therapies and clinical evolution of patients—paper data

4. data on drug orders—mix of digital and paper data

5. protocols, guidelines, interviews with clinicians.

In the following, for each of the 5 data sources, we specify the categories of data collected and the period observed.

In regards to data coming from Prosafe, we collected the records of daily admissions/discharges in the ward and patients' clinical condition severity at admission along five years (from January 2011 to April 2016, summing up to 2502 hospitalization). Data from Prosafe belong to the following categories:

- administrative data of the patient: biographical data, medical record number and admission/discharge dates from the hospital and ICU

- clinical conditions on admission, including any diseases, cancers and types of diabetes

- scores of patient severity, calculated from the values of important vital parameters, including heart rate, systolic blood pressure, white blood count (WBC), platelets and bilirubin

- information on any patient support devices

- information on any surgical procedures.

Prosafe data represent only a snapshot of the patient's condition on arrival in the ward and on discharge; daily information is therefore missing in this database.

In regards to the microbiology tests, we use a structured set of data coming from the microbiology laboratory. These data refer to a six-year period (from January 2010 to February 2016) and are organized in 3 different.csv files. They contain the list of patients with the date tests are requested, the sequence of tests done on the same patient according to the evolution of clinical

conditions, test results, and, for each test with a positive result, a rank of antibiotics, personalized patient by patient, which identifies the most expected effective treatments (antibiograms).

In regards to paper information, medical records for patients admitted in 2015 and 2016, numbered and organized into files, are in the paper archives of the ICU. For earlier hospitalizations, the records are in an archive located in a neighboring municipality and can be consulted one at a time after formal requests and time-consuming bureaucratic procedures. Each medical record is associated with the hospitalization of a patient and contains for each day of hospitalization a record compiled by clinicians and a record compiled by nurses. For each patient and each day of hospitalization, the nursing record is updated manually every two hours and contains information such as hemodynamic parameters, ventilation mode (if any), and Glascow Coma Scale score. For each patient and each day of hospitalization, the medical record is also updated manually every two hours and contains drug prescriptions, together with the mode drugs are administered (e.g. intravenous, subcutaneous, intramuscular, fluid therapy, oral).

In regards to drug orders, some of them are done automatically (ESTAR, [32]), while other are filled manually. In this study, we digitized drug orders and medical records corresponding to the 69 admissions (66 patients) occurred in June 2016. It is important to note that, unlike reported on other studies [33], in our case the database population was not automatically done and resulted in (i) burden on caregivers and (ii) interference with their workload. All the staff at the ICU department was willing to provide us with information and transfer knowledge about the entire processes ruling the ward, often outside of their work hours. However, to contain the impact of data collection on their daily activities, the acquisition of patient data from paper medical records, and data related to drug orders was limited to a time horizon of 1 month.

In regards to the acquisition of knowledge about the process ruling the system dynamics in the ward, we collected information from medical guidelines and protocols [34], as well as from interviews with clinicians, nurses and caregivers. Specifically, according to guidelines, when clinicians suspect that an infection is in progress, they issue a microbiology laboratory test and start an empirical treatment until the results from the lab are returned. Empirical treatment (referred to as ET in the following) is broad spectrum and covers the most likely microorganisms. As soon as laboratory tests identify the pathogen, a target treatment (referred to as TT in the following) begins. The ongoing treatment is continuously monitored and possibly interrupted (de-escalation) in case of clinical deterioration or drug resistance.

The 5 data sources feed two datasets. The first, consisting of 2502 admissions, is the result of combining, in the same time horizon, data from Prosafe and the microbiology laboratory. This dataset is used in Section 4.2 for data analysis and specifically, for patient stratification and prediction of Los and infection outbreak. The second dataset instead, is composed of 66 patients, corresponding to 69 admissions, and is obtained by cross-referencing manually collected data on therapies and drug orders. This dataset is used to recover the drug demand of a specific month and to obtain a first comparison between the optimized solution and the real case. The related results are shown in Section 6.2.

## 4.2 Data analysis

The information described above and coming from Prosafe and the microbiology laboratory have been pre-processed through the R language and yielded a table whose rows are the patients and columns the *features*, i.e. patient information, divided by category: administrative data, vital parameters, clinical conditions at admission, devices, microbiology test results. Admissions can be of three types: elective surgeries, urgent surgeries, medical. Specifically, in

the data collection period, of the 2502 admissions, 1852 (74.02%) were elective surgeries, 341 (13.63%) urgent surgeries, and 308 (12.31%) clinical. A more in-depth analysis, stratified over the years, shows how this relationship between the three types remains more or less constant until 2014, with an even more pronounced prevalence of elective surgical admissions in 2014 and 2016 and evidence of a reverse trend in 2015, when they decrease to make room for hospitalizations of medical type, which result in almost a doubling. Data from early 2016 show an almost equal percentage of medical and urgent surgery admissions. In contrast, previous years show an increase in admissions of the emergency surgical type at the expense of the medical type. The data reflect two events that occurred in the ICU during the observation period: (i) the expansion of the ward from 4 to 6, and finally to 8 beds; (ii) the change in the specialty of elective surgical patients. In fact, in the hospital considered there are several ICUs organized by specialty. In the period of observation, 209 deaths occurred, mostly in the first days of hospitalization. Specifically, looking at the data stratified by type of admission, the number of deaths occurred are respectively 50, 51, and 108 for elective surgeries, urgent surgeries, and medical, corresponding respectively to death rates equal to 2.7%, 15.0%, and 35.0% for the three types. It clearly emerges that the mortality rate is significantly higher for medical patients than for the other two categories, in accordance with the severity of their clinical conditions.

In the observation period, over the years, the percentage of patients with at least one request to the microbiology laboratory ranges from a minimum of 17.42% in 2016 to a maximum of 32.24% in 2015. In the same period, the percentage of positive results ranges from a minimum of 58.24% in 2014 to a maximum of 70.37% in 2016.

**4.2.1 Patient stratification.** A correct estimate of the patient LoS plays a pivotal role in the ward management since the LoS is strictly correlated to clinical condition severity and usage of human and material resources. We conducted a preliminary analysis in which patients are stratified according to the type of admissions and several metrics on the LoS are observed. The results of such an analysis are given in Table 1. Specifically, the columns report separately for each year and admission type, respectively the year considered, the type of

**Table 1. Statistics on LoS stratified by year and admission type.**

| Year | AdmType | Min | Max | avg | median | StdDev | Variance | 95% | NHosp | NDeath |
|------|---------|-----|-----|-----|--------|--------|----------|-----|-------|--------|
| 2011 | ElectiveSurgery | 1 | 44 | 2.84 | 2 | 3.91 | 15.28 | 8 | 248 | 16 |
| 2012 | ElectiveSurgery | 1 | 57 | 2.90 | 2 | 4.75 | 22.56 | 6 | 258 | 13 |
| 2013 | ElectiveSurgery | 1 | 18 | 2.37 | 2 | 2.41 | 5.82 | 6 | 275 | 7 |
| 2014 | ElectiveSurgery | 1 | 10 | 1.90 | 1 | 1.38 | 1.89 | 5 | 537 | 3 |
| 2015 | ElectiveSurgery | 1 | 43 | 2.60 | 1 | 3.81 | 14.55 | 7 | 411 | 8 |
| 2016 | ElectiveSurgery | 1 | 11 | 2.11 | 1 | 1.74 | 3.02 | 5 | 123 | 3 |
| 2011 | UrgentSurgery | 1 | 82 | 7.78 | 4 | 13.04 | 170.18 | 21 | 54 | 16 |
| 2012 | UrgentSurgery | 1 | 54 | 5.36 | 3 | 7.65 | 58.57 | 16 | 67 | 6 |
| 2013 | UrgentSurgery | 1 | 64 | 7.24 | 3 | 10.79 | 116.42 | 25 | 51 | 6 |
| 2014 | UrgentSurgery | 1 | 37 | 5.55 | 3 | 6.61 | 43.66 | 16 | 85 | 9 |
| 2015 | UrgentSurgery | 1 | 43 | 5.10 | 3 | 6.74 | 45.41 | 18 | 68 | 11 |
| 2016 | UrgentSurgery | 1 | 17 | 7.73 | 8 | 6.16 | 37.92 | 17 | 16 | 3 |
| 2011 | Medical | 1 | 39 | 7.56 | 5 | 7.91 | 62.52 | 19 | 45 | 18 |
| 2012 | Medical | 1 | 37 | 9.28 | 7 | 8.19 | 67.12 | 22 | 36 | 9 |
| 2013 | Medical | 1 | 78 | 8.90 | 5 | 13.78 | 189.94 | 34 | 42 | 13 |
| 2014 | Medical | 1 | 27 | 5.84 | 3 | 6.20 | 38.43 | 19 | 69 | 30 |
| 2015 | Medical | 1 | 38 | 7.38 | 6 | 7.11 | 50.48 | 26 | 100 | 32 |
| 2016 | Medical | 1 | 28 | 7.69 | 5 | 7.36 | 54.10 | 22 | 16 | 6 |

admission, the minimum, maximum, average and median values of the LoS in the class, standard deviation and variance, the 95 percentile, the number of hospitalizations (NHosp) and deaths (NDeath).

As the table clearly shows, the average and median LoSs are lower for elective surgical patients than for the others, as they typically have less severe conditions. For almost all the groups of patients observed (i.e., for each table row), there is a significant difference between the minimum and maximum value of LoS, and the standard deviation is very high. The results thus clearly show that a patient stratification based exclusively on type of admission provides only a poor estimation of the LoS. Motivated by these considerations, we investigated the possibility of classifying patients according to multiple features. The features to consider have been extensively discussed with clinicians and two alternative classifications were identified as the most promising ones among possible classifications: a 3-feature classification, and a 4-feature classification. In both the classifications, the following features have been considered: type of admission (elective surgery, urgent surgery, medical), reason of admission (medium care intensity, high care intensity), and ventilation (yes/no). In the 4-feature classification, the presence of an infection at admission (yes/no) is also considered. The number of classes in which patients are organized clearly depends on the number of features and the number of values each feature can assume. Thus, the number of classes is 3 for the 1-feature classification, 12 in the 3-feature classification and 24 in the 4-feature classification. Each class is then stratified by year since, as observed above, during the observation period, the department has undergone reorganization phases that have modified both the number and type of patients admitted. The quality of the classifications has been measured via a weighted standard deviation computed as the average of the standard deviations of each class of patient in each year, weighted by the number of hospitalizations corresponding to the class considered. Such a quality indicator has been computed also excluding the deaths. Table 2 reports for each classification the features considered, their number and the weighted standard deviation including and excluding the deaths. The table clearly shows that in all the classifications the accuracy improves significantly excluding the deaths. The 4-feature classification, as expected, performs better than the others, but with respect to the 3-feature classification a limited quality improvement is observed. In addition, considering a high number of classes, it may happen that some classes are made up of few patients, thus affecting the applicability of the results obtained. For these reasons, in agreement with the clinicians, it was concluded that the 3-feature classification represents a good compromise between accuracy and significance. The demand generator described in Section 4.3 is based therefore on the 3-feature classification of the patients.

The probability of death (Death Prob) and the probability that a patient arriving at ICUs belong to a certain class (Class Prob) have been computed for each class and they are reported in Table 3.

The 3-feature classification has also been the basis of a first attempt to predict LoS and infection outbreak. The preliminary results obtained are briefly described in the following.

**4.2.2 LoS prediction.** An accurate prediction of LoS is critically important for the department management. In this section, we report preliminary results obtained with a set of prediction tools used to predict the LoS of patients belonging to a certain class. Specifically, the

**Table 2. Classification quality—With and without deaths.**

| Class | NFeatures | WeightedSD | WeightedSD_noDeaths |
|---|---|---|---|
| AdmType_AdmRea_Vent_InfAdm | 4 | 3.43 | 2.55 |
| AdmType_AdmRea_Vent | 3 | 3.67 | 2.73 |
| AdmType | 1 | 4.30 | 3.27 |

**Table 3. Statistics on the 3-feature classification.**

| Class | | | | | | | | |
|---|---|---|---|---|---|---|---|---|
| AdmissionType | CareIntensity | Vent | NHosp | NOutliers | Death Prob | Class Prob | LoS estimate 95% | LoS estimate |
| ElectiveSurgery | Medium | No | 872 | 34 | 0.46% | 34.99% | 1-3 | 1-4 |
| ElectiveSurgery | Medium | Yes | 772 | 27 | 1.17% | 30.94% | 1-5 | 1-6 |
| ElectiveSurgery | High | No | 32 | 1 | 12.50% | 1.28% | 1-6 | 1-7 |
| ElectiveSurgery | High | Yes | 175 | 8 | 18.86% | 7.01% | 1-8 | 1-10 |
| UrgentSurgery | Medium | No | 62 | 3 | 3.23% | 2.48% | 1-4 | 1-6 |
| UrgentSurgery | Medium | Yes | 95 | 4 | 6.32% | 3.81% | 1-6 | 1-6 |
| UrgentSurgery | High | No | 21 | 1 | 9.52% | 0.84% | 1-7 | 1-14 |
| UrgentSurgery | High | Yes | 161 | 6 | 24.84% | 6.45% | 1-14 | 1-18 |
| Medical | Medium | No | 52 | 3 | 9.62% | 2.08% | 1-10 | 1-11 |
| Medical | Medium | Yes | 17 | 1 | 5.88% | 0.68% | 1-9 | 1-9 |
| Medical | High | No | 34 | 2 | 35.29% | 1.36% | 1-8 | 1-8 |
| Medical | High | Yes | 201 | 6 | 43.28% | 8.06% | 1-18 | 1-27 |

3-feature classification described in the previous section is assumed to stratify patients. Data are organized in a table where rows correspond to samples (the patients) and columns to features (data from Prosafe along five years). The target value is the LoS. A pre-processing phase is required to: (*i*) transform categorical features into binary features; (*ii*) scale feature values; and (*iii*) identify correlation between data. The tools used are: SVR with linear kernel for prediction, $k$-fold cross-validation with $k = 3$ for validation, and Relieff [35] for features ranking. The Mean Absolute Error (MAE) is used to measure solution quality. For each patient class, we use 4/5 of the data for the training phase and 1/5 for the test phase. Only classes with at least 50 patients have been considered. As an example, Table 4 reports the MAE value (averaged over the folds) obtained for the best ten configurations of parameters $\epsilon$ and $C$ used in the validation phase for the largest class in the database (872 patients), i.e. elective surgery, medium care intensity, no ventilation.

The accuracy obtained in test phase for the best performing parameter configuration (first row in the table) is equal to 1.16 days. The feature selection reveals that with only 6 of the 143 features characterizing patients in this class the accuracy can even improve to 1 days. Specifically, the 6 key features seem to be the following: average blood pressure, digestive tract perforation, surgical department from which the patient comes, metabolic failure, vascular abdominal surgery, and type-2 diabetes.

**Table 4. Results of LoS prediction for the ElectiveSurgery_MediumCareIntensity_NoVentilation class—SVR with 3-fold cross-validation.**

| $\epsilon$ | $C$ | accuracy |
|---|---|---|
| 0.0625 | 0.0625 | 0.989 |
| 0.125 | 0.0625 | 1.004 |
| 0.0625 | 0.125 | 1.008 |
| 0.0625 | 0.25 | 1.010 |
| 0.0625 | 0.5 | 1.011 |
| 0.0625 | 16 | 1.012 |
| 0.0625 | 4 | 1.012 |
| 0.0625 | 2 | 1.012 |
| 0.0625 | 8 | 1.012 |
| 0.0625 | 1 | 1.012 |

**Table 5. An overview of LoS prediction.** Each class with at least 50 patients is considered separately as a dataset, CI in the name of the dataset stands for Care Intensity. Accuracy is computed as average MAE—expressed in number of days. Feature selection can be enabled (FeatSel) or not (NoFeatSel), column Class reports the LoS range provided by classification, outliers excluded.

| Dataset | MAE | | | NFeatures | |
|---|---|---|---|---|---|
| | NoFeatSel | FeatSel | Class | NoFeatSel | FeatSel |
| ElectiveSurgery_MediumCI_NoVent | 1.16 | 1.00 | [1, 4] | 143 | 6 |
| ElectiveSurgery_MediumCI_Vent | 1.46 | 1.75 | [1, 6] | 135 | 4 |
| ElectiveSurgery_HighCI_Vent | 5.15 | 4.46 | [1, 10] | 145 | 6 |
| UrgentSurgery_HighCI_Vent | 5.68 | 5.56 | [1, 18] | 154 | 7 |
| UrgentSurgery_MediumCI_NoVent | 1.43 | 1.30 | [1, 6] | 102 | 7 |
| UrgentSurgery_MediumCI_Vent | 1.83 | 2.08 | [1, 6] | 117 | 19 |
| Medical_MediumCI_NoVent | 3.00 | 3.40 | [1, 11] | 90 | 7 |
| Medical_HighCI_Vent | 7.45 | 6.20 | [1, 27] | 155 | 8 |

Table 5 reports for the 8 over 12 classes considered, the accuracy obtained (MAE) in test phase averaged over the folds without and with feature selection. Then, the Los range provided by classification is reported in brackets in column Class, and finally, the last two columns give the number of features used without and with features selection.

**4.2.3 Prediction of infection outbreak.** As discussed above, when the clinician suspects an infection, an empirical therapy is carried on until the results from the lab arrive and a target therapy can then start. Usually, test results arrive after 3-7 days, so having an accurate prediction of any microorganisms present would be of paramount importance to define the target therapy at an early stage and also to limit drug resistance, which is particularly dangerous in fragile patients. Such an objective, although very challenging, is unfortunately beyond the scope of our study mainly due to the lack of data. What we did, therefore, was to carry out a study on the prediction of the presence of microorganisms (yes/no answer). To this purpose, data from microbiology were used, but the observation period was restricted to the period before the events that changed the number and type of patients admitted (2011-2014). It was decided to use the oldest part of the database both because it is a longer period with respect to the more recent part, and to avoid the effects of the transitional period that necessarily follows an event. The total number of samples thus decreased to 653, of which 553 used in the training phase and 100 in the test phase. In addition, patients are not stratified to avoid small and unusable classes. In this section, we report very preliminary results obtained with: SVM for prediction, $k$-fold cross-validation with $k = 3$ for validation, and Relieff [35] for features ranking. The mean of sensibility (accuracy in predicting the "yes" label) and specificity (accuracy in predicting the "no" label) is used to measure solution quality.

Table 6 reports the accuracy value, averaged over the folds, obtained for the best ten configurations of parameters $\gamma$ and $C$. For the best performing configuration sensibility and specificity are respectively 1 and 0.85 and their mean (accuracy) is 92.5%. Feature selection allows to consider only 5 over the 213 features obtaining 0.91 and 0.80 respectively for sensibility and specificity with an accuracy equal to 85% in the test phase which, though preliminary, is a quite encouraging result.

**4.2.4 Extraction of empirical probability distributions.** From data collection, we extracted the following probability distributions which have been used to feed the demand generator:

- $pd_1$ probability that on a given day a certain number of incoming patients are admitted

- $pd_2$ probability that a certain patient belongs to a certain class

**Table 6. Results of infection prediction—SVM with 3-fold cross-validation.**

| $\gamma$ | $C$ | accuracy |
|---|---|---|
| 0.125 | 0.25 | 0.790 |
| 0.25 | 1 | 0.788 |
| 0.25 | 0.25 | 0.787 |
| 0.0625 | 0.03125 | 0.784 |
| 0.25 | 0.5 | 0.784 |
| 0.125 | 0.125 | 0.783 |
| 0.0625 | 0.5 | 0.783 |
| 0.125 | 0.5 | 0.783 |
| 0.125 | 0.0625 | 0.780 |
| 0.0625 | 0.125 | 0.779 |

- $pd_3$ probability that the LoS of a certain patient belonging to a certain class is a certain number of days

- $pd_4$ probability that there is suspicion of infection for a certain patient belonging to a certain class and with a certain LoS, on a given day after admission

- $pd_5$ probability that for a certain patient (with given class and Los) new laboratory tests will be required after a certain number of days (or equivalently that the ongoing treatment has not been working for a certain number of days).

Specifically, Table 7 reports for each probability distribution the sources of data used to compute them.

While $pd_1$, $pd_2$, and $pd_3$ are at the core of the prediction of the ward occupancy, as illustrated in Fig 1, $pd_4$ and $pd_5$ rule the transition from a patient status to the next, for a given patient of a certain class with a certain LoS, as depicted in Fig 2.

## 4.3 Description and use of demand generator

The demand generator exploits the empirical probability distribution functions above introduced.

Regarding daily patient admission, $pd_1$ has been derived by the data recorded in the Prosafe database concerning admission and discharging date of each patient and patient death. In particular, for each day $t$, the following data can be extracted: i) number of admitted patients $in_t$; ii) number of patients $out_t$ who leave the ward, either discharged (whatever the cause) or because of their death; iii) bed occupancy level $bol^t \in \{0..B\}$, where $B$ is the number of beds the ward is equipped with in that period. Clearly, $bol_t = bol_{t-1} - out_t + in_t$ holds. The number of empty beds is given by $vacancy_t = B - bol_{t-1} + out_t$. Then, $pd_1(i|v)$ returns the probability that

**Table 7. Data source for each empirical probability distribution.**

| ProbDistr | Prosafe | LabResults | Therapies | Drug Orders | Process Knowledge |
|---|---|---|---|---|---|
| $pd_1$ | ✓ | | | | |
| $pd_2$ | ✓ | | | | ✓ |
| $pd_3$ | ✓ | | | | ✓ |
| $pd_4$ | ✓ | ✓ | ✓ | ✓ | ✓ |
| $pd_5$ | ✓ | ✓ | ✓ | ✓ | ✓ |

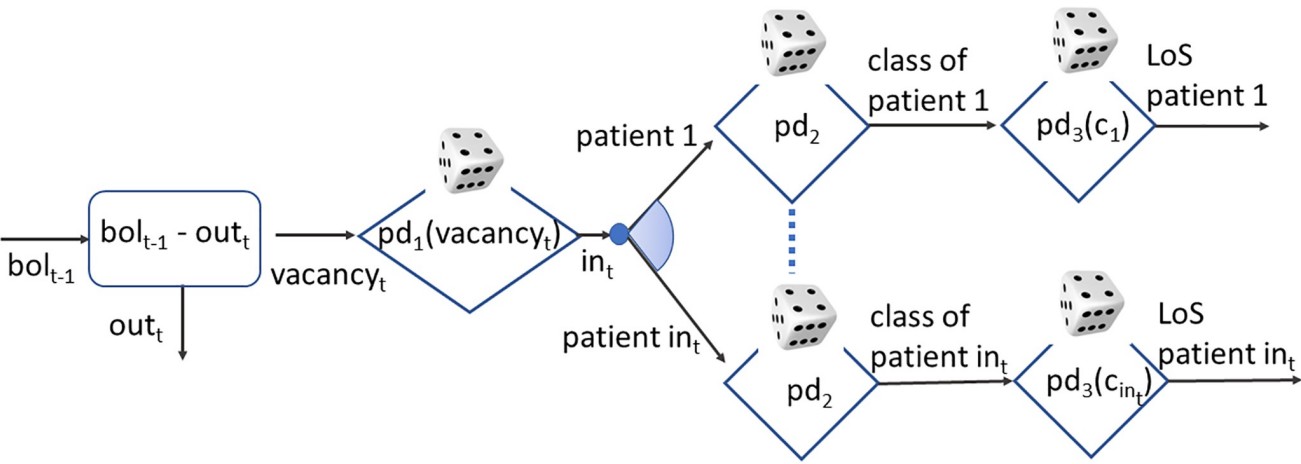

**Fig 1. Flow chart of the generation process of ward bed occupancy.**

$i$ patients are admitted given a vacancy of $v$. Such probability is computed as the ratio of the number of days $t$ in which $i$ patients were admitted, given $v$ empty beds, over the number of days when $v$ beds were available, i.e., $pd_1(i|v) = (\sum_{t\in T:vacancy_t=v}\gamma_i^t)/(\sum_{t\in T:vacancy_t=v}1)$ where $\gamma_i^t = 1$ if $in_t = i$, 0 else.

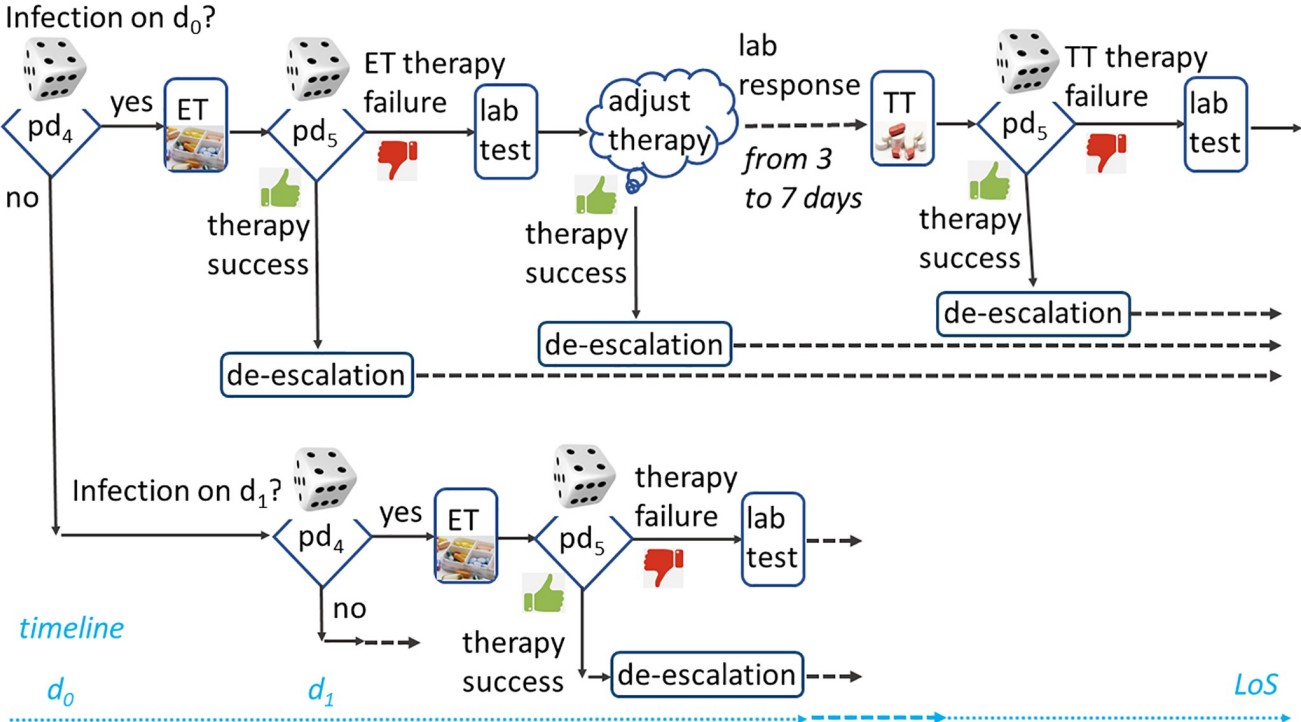

**Fig 2. A representation of (part of) a tree describing the evolution of a patient clinical conditions.** In the full tree, each decision node (such as infection onset date if ever, therapy outcome, or the time needed for the answer from the microbiology lab) will have as many descendants as the possible decision outcomes. Dice icons represent stochastic events (tree nodes), while dotted lines model time progress.

The other empirical probability distribution functions have been likewise computed, as the ratio of favourable cases over possible cases.

Fig 1 represents the flow chart of the patient generator process. It is intended to reproduce the patient flow through the ward, including the patient type. Each day $t$, the number of empty beds available after discharge is computed. Given the current vacancy, a certain number of new patients are admitted according to $pd_1$. Each new patient belongs to a certain class, according to $pd_2$, and has a certain LoS, according to $pd_3$, given that class.

The concept of *path* is very helpful in building an abstract representation of the ward as a system. The generator, for each patient class, defines a set of paths representing possible evolution of patients' clinical conditions. Indeed, the chain of events which characterize the evolution of the clinical conditions of a patient, can be modeled as a path along a tree, whose nodes correspond to stochastic events and branches to possible evolutions. Examples of events are the antibiogram laboratory response, the development of antibiotic resistance, the clinician's choice regarding the empirical therapy, and the patient's response to therapy. The generator also introduces fuzzy events that occur with a given probability and model a stochastic event (e.g., death) or the expertise of the clinician who, at any time, may decide to do a therapy de-escalation and a therapy switch. A treatment and its daily drug prescription are associated with each node of the tree. Individual patient prescriptions are summed up to yield the daily ward drug demand. A (partial) representation of a patient tree is provided in Fig 2. Different paths can be followed from admission on day $d_0$ to discharge on $d_0 + LoS$. The first stochastic event concerns the onset day of an infection, if ever. In such a case, the empirical therapy may fail or succeed (second stochastic event) and so on, as already discussed.

The generator is used in two different ways in our optimization-simulation approach. In both cases, it receives as input the current population of patients in the ward. In the first case, when it is used in the simulator, it evolves a patient's status, day by day, patient by patient, and it populates the ward with new patients according to the arrival process, thus revealing the actual demand incrementally. In the second case, the generator is used as a predictor to yield in one shot the drug demand for the coming planning period. The resulting demand is used to feed the optimization model.

## 4.4 Some hints on alternative approaches to uncertainty

We conclude Section 4 with a brief discussion on alternative approaches to handle demand uncertainty. Indeed, suppose you are facing an optimization model in which some problem data are uncertain, i.e. drug demand in our problem. If probability distribution function of random parameters is known, a straightforward approach is to compute their expected values and solve the model with respect to such values as it were deterministic. When the probabilistic description is missing, a similar option may be pursued by computing the most likely value of the random parameters by exploiting historical data sets or by relying upon experts knowledge. In all cases, results can be quite far from optimality, as the optimal solution of such a model typically does not correspond to the one providing the optimal expected value of the objective function, thus providing the decision maker with misleading directions.

A more realistic representation of the parameters variability exploited by stochastic optimization [36] involves computing a limited set of scenarios, corresponding to the different realizations of the random parameters. Each scenario is then weighted by a probability, and an optimization model is solved which optimizes the weighted cost function and whose constraints model each scenario. This option captures potential inter-dependencies among random parameters that may be difficult to represent analytically. However, some scenario aggregation techniques are often to be used to reduce the size of the resulting model. When

the probability distribution function of the parameters in unknown, one can resort to robust optimization [37]. In robust optimization, the uncertain domain is set based rather than being described by a probability function. For example, the uncertain parameters are allowed to continuously vary within an interval or in a discrete set. The total deviation from each nominal value of the uncertain parameters is constrained. A risk adverse attitude is implemented, in the sense that a solution is searched such that is feasible for any realization and it provides the best solution in the worst scenario.

In our framework, these options would give rise to the following. The expected values strategy would solve the mathematical model introduced in Section 5.1 in which the daily demand of each drug is the average demand in one day in that period. The current ward situation, in terms of hospitalized patients and their therapies, would be completely disregarded. Regarding scenario based stochastic optimization, each scenario could correspond to a possible patient path on the patient's tree, and its probability could be computed based on the previously introduced distribution functions. On regards to solution cost, rush orders should be included as second stage recourse actions (see [38]) to be taken in case of stock outs. Given the width of the time horizon the model encompasses and the many possible options along a path, the number of scenarios would likely rise to a number that makes this approach unpractical. A robust optimization approach could be realized by considering a tree for each patient rather than the patient's nominal path, that is much smaller than the full patient tree. Such a tree could be obtained from the nominal path by adding only one branch at each node, the one that leads to the most probable outcome different from the one followed by the nominal path at that decision node, and then proceeds along a unique trajectory with no other branches. The resulting structure is a particular binary tree for each patient, with one branch per level. The robust problem is solved with the constraint that at most a certain number of patients can deviate from their nominal path. These approaches could be worth exploring, and will the subject of further studies.

## 5 The optimization-simulation approach

We propose to exploit the knowledge on hospitalized patients and therapies in order to periodically set up an agenda of push orders that should provide baseline replenishment for the current period, which is intended to reduce the need for additional orders, day by day. The push order agenda, once computed, may be sent to and managed by the hospital pharmacy. Nurses would thus be aware of the order agenda and would be able to plan their time in advance as in periodic review policies, but they would be released from the decision-making process. Push orders are set by solving a deterministic MILP model with respect to a forecast demand scenario. The model (see Section 5.1) encompasses all the features in Section 3, as it aggregates drug orders over time, guarantees regular order quantities, handles limited storage and budget, and considers the most influential stakeholder perspective. The simulator then deploys the push agenda day by day, updates the inventory and issues rush and extra orders (i.e., pull orders) when the realized demand differs from the forecast. Rush orders are required to cope with stock outs, whereas extra orders restore safety stocks (see Section 5.2 for their estimate). At the end of each period, based on the status of current patients, another demand forecast spanning the next period is generated (Section 4.3) and forwarded to the mathematical model, together with the actual inventory levels. The process is thus iterated in a rolling horizon framework (Section 5.3). The main aforementioned components of the algorithm are detailed hereafter, and Fig 3 provides a graphical representation of the overall approach.

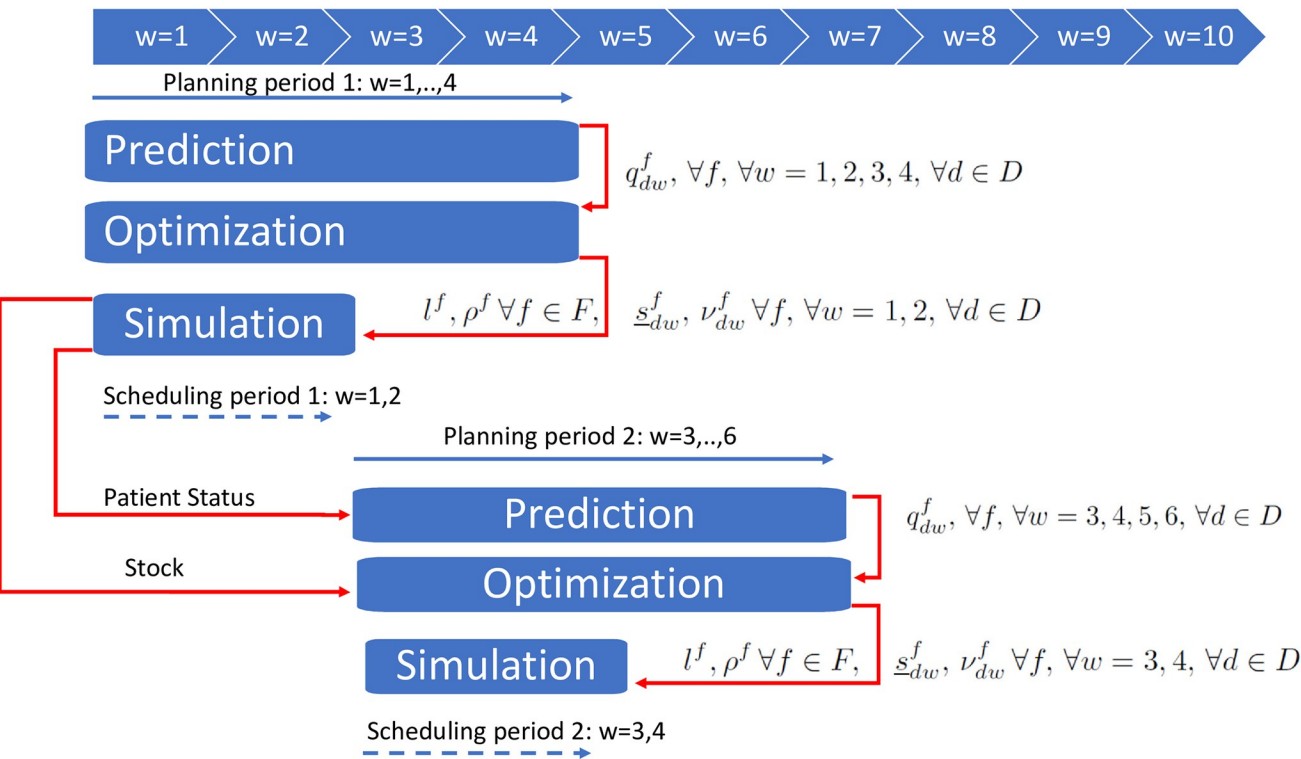

**Fig 3. System components and their integration.**

## 5.1 An MILP mathematical model

The mathematical model used in this study extends [39] in several directions. Given the demand for the current period, a set of drugs, and other parameters such as the inventory on the first day, the model determines, for each drug: (*i*) when a drug is to be ordered (reorder points) and (*ii*) the quantity to be ordered every time an order of this drug is triggered in the planning horizon (reorder lot). The main variables thus concern the stock level of each drug at the end of each day of the planning period, the order quantity of each drug, and the schedule of order events. The mathematical notation and the model variables are summarized in Tables 8 and 9.

The constraints describing the operations in the ward can be organized in the following blocks: (*i*) flow conservation constraints on the stock level of drugs, (*ii*) warehouse capacity constraints (dedicated and shared storage), (*iii*) budget constraints, (*iv*) constraints on the regularity of orders, (*v*) constraints on stakeholders' perspective, and (*vi*) constraints on the variable domain. The model is defined as follows:

$$\min \quad \varepsilon + M \sum_{d \in D, w \in W} v_{dw} + \bar{O} \sum_{f \in F} \bar{v}_{01}^f + \sum_{f \in F} \sum_{d \in D, w \in W} \frac{c^f}{B_{dw}} s_{dw}^f \tag{1}$$

$$s_{01}^f = l^f - q_{01}^f + U^f \bar{y}_{01}^f \bar{v}_{01}^f \quad \forall f \in F \tag{2}$$

$$s_{dw}^f = s_{d-1,w}^f - q_{dw}^f + U^f y_{d-1,w}^f \quad \forall f, \, \forall d \in D, d \neq 0, \, \forall w \geq 1 \tag{3}$$

**Table 8. Sets and parameters.**

| $F$ | set of drugs (indexed by $f$) |
|---|---|
| $G$ | set of drug groups (indexed by $g$)—drugs in the same group share the storage unit |
| $F_g \subseteq F$ | set of drugs in group $g \in G$ |
| $D = \{0, \ldots, 6\}$ | ordered set of days (indexed by $d$, 0 corresponds to Sunday, etc) |
| $W$ | set of weeks (indexed by $w$, with $w \geq 1$) |
| $q_{dw}^f$ | demand of drug $f$ on day $d$ of week $w$ (number of doses) |
| $U^f$ | number of doses in each box of drug $f$ |
| $c^f$ | cost of each dose of drug $f$ |
| $B_{dw}$ | maximum allowed stock monetary value on day $d$ of week $w$ |
| $I^f$ | doses of drug $f$ in the ward at time 0 |
| $\underline{s}_{dw}^f$ | safety stock of drug $f$ on day $d$ of week $w$ |
| $C^f$ | capacity of the storage unit dedicated to drug $f$—number of boxes |
| $\bar{C}^f$ | maximum number of drug $f$ boxes in the shared storage unit |
| $V^f$ | volume of a drug $f$ box in number of units in the shared storage unit |
| $\bar{V}_g$ | capacity of the shared storage unit for group $g$, expressed in number of units |
| $\Gamma^f = C^f + \bar{C}^f$ | upper bound on the number of boxes of drug $f$ in stock |
| $\bar{O}$ | cost of an urgent order |
| $\bar{y}_{01}^f$ | number of boxes of drug $f$ in rush order on day 0 week 1 |
| $\bar{v}$ | reference value for the level of inhomogeneity in orders |

$$s_{dw}^f = s_{6,w-1}^f - q_{dw}^f \quad \forall f \in F, \; d = 0, \; \forall w \geq 2 \tag{4}$$

$$s_{dw}^f \leq U^f \left( C^f + x_{dw}^f \right) \quad \forall f \in F, \; \forall d \in D, \; \forall w \in W \tag{5}$$

$$\sum_{f \in F_g} x_{dw}^f V^f \leq \bar{V}_g \quad \forall g \in G, \; \forall d \in D, \; \forall w \in W \tag{6}$$

$$(\beta_{dw}^f - 1) U^f + 1 \leq s_{dw}^f \quad \forall f \in F, \; \forall d \in D, \; \forall w \in W \tag{7}$$

**Table 9. Variables.**

| $s_{dw}^f$ | stock level of drug $f$ at the end of day $d$ of week $w$, expressed in number of doses |
|---|---|
| $y_{dw}^f$ | order quantity of drug $f$ on day $d$ of week $w$, expressed in number of boxes |
| $v_{dw}^f$ | equal to 1 when a standard order of drug $f$ occurs on day $d$ of week $w$; 0 otherwise |
| $v_{dw}$ | equal to 1 when a standard order occurs on day $d$ of week $w$; 0 otherwise |
| $\bar{v}_{01}^f$ | equal to 1 when an urgent order of drug $f$ occurs on day 0 of week 1; 0 otherwise |
| $\rho^f$ | order quantity of drug $f$, expressed in number of boxes |
| $x_{dw}^f$ | number of boxes of drug $f$ in the shared storage unit on day $d$ of week $w$ |
| $\alpha_{dw}^f$ | number of boxes of drug $f$ in the dedicated storage unit on day $d$ of week $w$ |
| $\beta_{dw}^f$ | number of boxes of drug $f$ in stock at the end of day $d$ of week $w$ |
| $\gamma_{dw}^f$ | equal to 1 if the storage unit dedicated to drug $f$ is full on day $d$ of week $w$ |
| $v_{\max}$ | maximum number of drugs in an order over the planning period |
| $v_{\min}$ | minimum number of drugs in an order over the planning period |
| $\varepsilon$ | maximum violation of the budget constraint |

$$s_{dw}^f \leq U^f \beta_{dw}^f \quad \forall f \in F,\ \forall d \in D,\ \forall w \in W \tag{8}$$

$$\beta_{dw}^f = \alpha_{dw}^f + x_{dw}^f \quad \forall f \in F,\ \forall d \in D,\ \forall w \in W \tag{9}$$

$$\alpha_{dw}^f \leq C^f \quad \forall f \in F,\ \forall d \in D,\ \forall w \in W \tag{10}$$

$$\gamma_{dw}^f \leq \alpha_{dw}^f / C^f \quad \forall f \in F,\ \forall d \in D,\ \forall w \in W \tag{11}$$

$$x_{dw}^f \leq \bar{C}^f \gamma_{dw}^f \quad \forall f \in F,\ \forall d \in D,\ \forall w \in W \tag{12}$$

$$\sum_{f \in F} c^f \left( s_{dw}^f - \underline{s}_{dw}^f - U^f \bar{y}_{01}^f \bar{v}_{01}^f \right) \leq B_{dw} + \varepsilon \quad d = 0,\ w = 1 \tag{13}$$

$$\sum_{f \in F} c^f \left( s_{dw}^f - \underline{s}_{dw}^f \right) \leq B_{dw} + \varepsilon \quad \forall d \in D,\ \forall w \in W \tag{14}$$

$$v_{dw} = 0 \quad d = 6,\ \forall w \in W \tag{15}$$

$$\rho^f \leq \Gamma^f \quad \forall f \in F \tag{16}$$

$$y_{dw}^f \leq v_{dw}^f \Gamma^f \quad \forall f \in F,\ \forall d \in D,\ \forall w \in W \tag{17}$$

$$y_{dw}^f \leq \rho^f \quad \forall f \in F,\ \forall d \in D,\ \forall w \in W \tag{18}$$

$$y_{dw}^f \geq \rho^f - (1 - v_{dw}^f)\Gamma^f \quad \forall f \in F,\ \forall d \in D,\ \forall w \in W \tag{19}$$

$$v_{dw}^f \leq v_{dw} \quad \forall f \in F,\ \forall d \in D,\ \forall w \in W \tag{20}$$

$$\sum_{f \in F} v_{dw}^f \leq v_{\max} \quad \forall d \in D,\ \forall w \in W \tag{21}$$

$$v_{\min} \leq \sum_{f \in F} v_{dw}^f + (1 - v_{dw})|F| \quad \forall d \in D,\ \forall w \in W \tag{22}$$

$$v_{\max} - v_{\min} \leq \bar{v} \tag{23}$$

$$\varepsilon \geq 0 \tag{24}$$

$$s_{dw}^f \geq \underline{s}_{dw}^f \quad \forall f \in F,\ \forall d \in D,\ \forall w \in W \tag{25}$$

$$y_{dw}^f, x_{dw}^f, \alpha_{dw}^f, \beta_{dw}^f \in Z^+ \quad \forall f \in F,\ \forall d \in D,\ \forall w \in W \tag{26}$$

$$\gamma_{dw}^{f} \in \{0,1\} \quad \forall f \in F, \ \forall d \in D, \ \forall w \in W \tag{27}$$

$$\upsilon_{dw} \in \{0,1\} \quad \forall d \in D, \ \forall w \in W \tag{28}$$

$$\upsilon_{dw}^{f} \in \{0,1\} \quad \forall f \in F, \ \forall d \in D, \ \forall w \in W \tag{29}$$

$$\bar{\upsilon}_{01}^{f} \in \{0,1\} \quad \forall f \in F \tag{30}$$

We first describe the constraints according to the block structure mentioned above, and then we come back to the objective function. Constraints (2) to (4) are classical flow conservation constraints regulating the stock level for each drug depending on the day considered, i.e., on the first day of the planning horizon (2), on week days (3), and on Sundays when no order is received (4). They guarantee that for each drug and for each day, the stock level of that drug at the end of the day is equal to the stock level at the end of the previous day plus the quantity of drug that possibly arrives in the ward on that day minus the consumption of that drug for that day. These constraints mirror the weak lot-sizing formulation [40] and could be tightened, as suggested in [41]. In a rolling-horizon framework, we cannot assume that the initial quantity of drugs in stock covers the demand on the first day of the current planning period, as opposed to using the model in one shot, as it happens in a single planning period. To overcome the potential infeasibility, we equipped the model with the possibility of triggering an urgent order on the first day of each planning period. Drugs ordered urgently are indeed available in the ward within two hours from order and thus can be promptly used on same day on which the order was made. For each drug $f$, the number of boxes $\bar{y}_{01}^{f}$ required to cover a possible stock out on the first day of the planning period can be a priori computed as follows:

$$\bar{y}_{01}^{f} = \max\left\{0, \left\lceil \frac{q_{01}^{f} + \underline{s}_{01}^{f} - l^{f}}{U_{f}} \right\rceil \right\} \quad \forall f \in F.$$

Thus, constraints (2) consider the possibility that an urgent order relative to drug $f$ is triggered (variable $\bar{\upsilon}_{01}^{f}$ equal to one) to receive a number of boxes $\bar{y}_{01}^{f}$ of drug $f$ sufficient to cover the demand. Quantity $\bar{y}_{01}^{f}$ is expressed in boxes, as orders occur in boxes, consistent with (3). If the demand is assumed to be known in advance, urgent orders later than day 1 lead to more expensive solutions, so we disregarded them. However, should urgent orders be used as a tool to avoid other kinds of infeasibilities, the model can be extended to take them into consideration.

The block of constraints (5) to (12) allows the management of dedicated and shared storage spaces. Specifically, constraints (5) and (6) impose that on each day, the quantity of drugs stored in dedicated, as well as in shared, spaces should not exceed the capacity. Capacity is expressed in number of boxes or in volume according to the type of storage considered, and proper conversions are done when required. In addition, constraints (7) to (12) guarantee, for each drug $f$, that the common storage space can be used only when the shared space is full. These constraints, besides reflecting ward common practice, help reduce solution symmetries: constraints (7) and (8) allow the computation of the number of boxes $\beta_{dw}^{f}$ in stock on day $d$ of week $w$ for drug $f$; $\beta_{dw}^{f} = \lceil s_{dw}^{f}/U_{f} \rceil$ with (7) and (8) linearizing such condition, and impose that $\beta_{dw}^{f}$ be equal to the minimum number of boxes that can contain $s_{dw}^{f}$ doses. Equivalently, $\beta_{dw}^{f} - 1$ boxes would not be sufficient to contain the doses in stock, whereas $\beta_{dw}^{f}$ does. Constraints (9), for each $f$, $d$, $w$, guarantee that the number of boxes in stock, namely, $\beta_{dw}^{f}$, is equal

to the sum of the number of boxes stored in the dedicated space, namely $\alpha_{dw}^f$, and the number of boxes stored in the shared storage space, namely $x_{dw}^f$. Constraints (10), separately for each drug $f$, impose an upper bound on the number of boxes of drug $f$ that can be stored on each day in the storage space dedicated to $f$. For each $f$, $d$, $w$, constraints (11) guarantee that the binary variable $\gamma_{dw}^f$ is fixed to zero whenever the storage space dedicated to $f$ is not full (case $\alpha_{dw}^f < C^f$); such a constraint, jointly with constraints (12), imposes that in this case, drug $f$ cannot be stored in the common space, i.e., $x_{dw}^f = 0$. Vice versa, when the dedicated storage space is full (case $\alpha_{dw}^f = C^f$), the binary variable $\gamma_{dw}^f$ can assume value one, and consequently, drug $f$ can be stored in the common space.

Constraints (13) and (14) allow the soft violation of budget constraints penalizing the violation in the objective function. Specifically, the cost of drugs stocked at the end of a day, excluding the cost of the safety stock and the cost of drugs that possibly arrive in the ward as a consequence of an urgent order on the first day of the planning period, can exceed the daily budget $B_{dw}$ of at most $\varepsilon$. Variable $\varepsilon$ thus represents the maximum violation of the budget constraints over the entire planning period, and it is minimized in the objective function. Even if, as mentioned, the minimization of the inventory costs is usually not pursued in a ward, the constraints used to control the daily budget are used as a means to prevent over stocking, which represents an undesirable event for the Hospital Management.

The block of constraints (15) to (20) controls order regularity. Specifically, these guarantee that for each drug, the same lot size is used every time such drug is ordered. In addition, they manage the order event variables, assuring that the variable related to an order event on a given day is equal to one, if at least one drug is ordered on that day.

The block of constraints (21) to (23) concerns nurses' perspective. Nurses play a prominent role among stakeholders because taking into consideration their needs positively affects the quality of the solution even from the point of view of the two other stakeholders. Specifically, nurses in charge of order management, besides minimizing the number of order events, ask for homogeneous orders in terms of the number of drugs in each order to better plan their activities. Indeed, the time required to store drugs into cabinets and update the system data (so far, on paper) mainly depends on the number of different drugs involved in the order rather than on the number of boxes, so they would like to keep this number steady. For these reasons, variable $v_{\max}$ in constraints (21) measures the maximum number of different drugs in an order over the planning period, whereas variable $v_{\min}$ represents the minimum number of different drugs in an order over the planning period. Then, in constraint (23), the difference between such maximum and minimum values is constrained to not exceed the reference value $\bar{v}$. In particular, the reference value is computed as follows:

$$\bar{v} = \left\lceil \sum_{i \in I} (v_{\max}^{(i)} - v_{\min}^{(i)}) / |I| \right\rceil - t_l$$

where $I$ is a set of instances used in the training phase and is considered a representative of the reality in the ward, $t_l$ is the threshold value corresponding to a low level of flexibility, and $v_{\max}^{(i)}$ and $v_{\min}^{(i)}$ represent respectively the maximum and the minimum number of drugs in an order over the planning period when the model is run on instance $i \in I$.

Finally, constraints (24) to (30) define the variable domain, and specifically constraints (25) impose that stock levels never decrease under the safety stock levels.

With regard to the objective function, it allows the hierarchical optimization of the following criteria: (*i*) minimization of budget violation, (*ii*) minimization of the number of urgent orders, (*iii*) minimization of the number of regular order events, and (*iv*) minimization of the

monetary value of drugs in stock. Constant $M$ is thus a weight properly set to reflect the hierarchy of the criteria involved. As explained above, the first two criteria, i.e., minimization of budget violation and minimization of urgent orders, are introduced to prevent infeasibilities that might occur when the model is used in a rolling horizon framework.

In conclusion, model (1)–(30) differs from that in [39] because (*i*) it allows urgent orders during the first day of the planning period, (*ii*) it controls the daily monetary value of the drugs in stock via soft constraints, (*iii*) it introduces safety stocks, and (*iv*) it prioritizes a dedicated storage space over a shared space for each drug.

## 5.2 Estimating the safety stock levels

For each drug *f*, a safety stock threshold $\underline{s}^f$ must be set and a pull order will be triggered if the inventory falls below it. The safety stock is intended to hedge from drug demand fluctuations during the lead time (equal to one day for regular orders in our setting). As we cannot rely on any statistical distribution of drug demand, we propose two alternative policies to set up the safety stock, both of which are based on our demand generator. The two policies will be referred to respectively as *basic* and *knapsack*. Specifically, we propose a two-step procedure: the basic policy is based only on the first step of the procedure, whereas the knapsack policy comprises both steps of the procedure. The detailed description of these two steps follows. Assume that $B_{ss}$ is the component of the budget *B* devoted to the safety stock. In the first step, for each drug, the quantity required to treat one patient for one day is set aside. We call this set the basic safety stock, and as mentioned above, it is the safety stock used in basic policy. $B_{ss}$ is thus decreased by the cost of this first basic stock of drugs. Let $\bar{B}_{ss}$ denote the residual budget. In the second step, the demand generator is run over a (very long) time horizon made of $H_P$ days. Then, for each drug, the empirical probability distribution on daily consumption is built by collecting the possible consumption values—usually integer multiples of a posology, i.e., what is required for one treatment—and then considering for each quantity the ratio of the number of days in which this quantity has been consumed over $H_P$ as its mass probability. Let $Val^f = \{\delta_i^f\}$ denote the set of (normalized with respect to the single posology amount) values with a positive probability, and let $Prob^f = \{\pi_i^f\}$ be the associated empirical probabilities. As an example, for a drug *f*, the posology can be equal to 4 doses, and the most frequent daily consumption values could be 8, 16, and 20, corresponding respectively to 2, 4, and 5 posologies of that drug. In such a case, $Val^f$ = {8, 16, 20} and set $Prob^f$ would contain the relative probabilities, with $Prob^f$ = {0.4, 0.2, 0.2}, meaning that a consumption of two posologies occurs with a probability equal to 0.4 whereas 4 and 5 posologies both occur with a probability equals to 0.2.

The following binary knapsack-like model is then solved

$$max\left\{\sum_{f \in F} \sum_{\pi_i^f \in Prob^f} \pi_i^f v_i^f \ \ s.t. \ \sum_{f \in F} \sum_{\delta_i^f \in Val^f} c^f \delta_i^f v_i^f \leq \bar{B}_{ss}, \ v_i^f \in \{0, 1\}\right\}$$

to yield $\underline{s}_f = \sum_{\delta_i^f \in Val^f} \delta_i^f v_i^f$ for each $f \in F$.

Note that more than one element in a set $Val^f$ may be chosen. This accounts for the fact that in our strategy, safety stocks should ideally be sufficient until the next push order of the drug. We call this enlarged set the knapsack-based safety stock, and it is the stock used in the *knapsack* policy. These two policies, *basic* and *knapsack*, will be experimentally evaluated (see Section 6).

In some sense, we might state that the knapsack policy is therapy oriented, whereas the basic one is drug oriented. In terms of the kind of information required to set up the two policies, the basic policy seems to be guided by the same information necessary to describe drug

consumption as a Monte Carlo process, whereas the knapsack policy reflects the drug consumption obtained by the demand generator.

## 5.3 A rolling horizon approach

The three system components, i.e., the demand predictor, the optimizer, and the simulator, are integrated within a rolling horizon framework. In short, the agenda of push orders is iteratively planned for the current period (the planning period) by solving the MILP model, but the decisions are implemented just for a shorter period, i.e., the scheduling period that covers only the first part of the planning period. More formally, consider a time horizon of $H$ days $T^H = \{1, .., H\}$ and two integer values, $\Omega$ and $\omega$, denoting respectively the length of the planning period and the length of the scheduling one, with $\omega < \Omega < H$ such that $H$, $\Omega$ and $\omega$, are integer multiples of 7, and $H$ is an integer multiple of $\omega$. At each iteration $it = 2, .., H/\omega$, we have just scheduled and processed the orders for the last $\omega(it - 1)$ days, i.e., we have scheduled the push orders and set up the pull orders whenever the actual demand required it, and we are ready to compute the push order agenda for the next $\Omega$ days. Thus, push orders are planned in the current planning period $T^\Omega$, going from day $\omega(it - 1)$ to day $\omega(it - 1) + \Omega - 1$, while they are implemented just for the first $\omega$ days, i.e., the scheduling period $T^\omega = \{\omega(it - 1), ..., \omega\, it - 1\}$.

Fig 3 exemplifies the process for $\Omega = 28$ and $\omega = 14$, depicting the information flow from the predictor, through the optimizer, to the simulator. More formally, let $P(it)$ and $Stock^f(it)$ denote the patient population hospitalized on the first day of $T^\Omega$ and the inventory level of drug $f$ on the same day, respectively. Note that the status associated with each hospitalized patient in $P(it)$ on that day corresponds to one state along the flowchart discussed in Section 4.3 for the associated patient class.

The demand is forecast for each day in $T^\Omega$ by launching the demand generator given $P(it)$. This process yields a demand matrix $Q^{it}$ that contains the demand of each drug $f \in F$ for each day in $T^\Omega$. Then, the MILP model is solved over $T^\Omega$ with respect to $Stock^f(it)$ and $Q^{it}$, which represent respectively the level of stock for each drug at the end of the previous scheduling period, namely, $l^f \forall f \in F$, and the demand for the current planning period, namely, $q^f_{dw} \forall f \in F, \forall w, d \in T^\Omega$. According to what has been discussed above, only the orders in $T^\omega$ are passed to the simulator forming the pre-scheduled agenda of the push orders for the current period. A brief sketch of the methodology proposed is given in Fig 4 as a pseudocode.

The rationale of the approach is that the MILP model requires a sufficiently large horizon to search for regularity. However, the more that we move forward in time within the period,

```
begin
    it=1; \\ iteration counter
    t=7(it-1); \\ first day in the current periods (planning and scheduling)
    T^Ω = {t,...,t + Ω − 1} \\ the planning period
    T^ω = {t,...,t + ω − 1} \\ the scheduling period
    while (T^Ω ⊆ T^H) do \\ still in the time horizon
        Q(it) = Prediction(T^Ω, P(it));
        PushAgenda(it) = Optimization(T^Ω, Q(it), Stock^f(it) ∀f);
        Deploy(it) = Simulation(T^ω, PushAgenda(it));
        it++;
    end while
end
```

**Fig 4. Pseudocode of the proposed approach.**

i.e., as we move ahead from the first day, the less reliable the demand forecast becomes. For this reason, the solution is inserted in the push agenda only for the first $\omega$ days.

The agenda is returned to the simulator, and this reveals the actual demand day by day, by a call to the demand generator one day at a time. Note that the patient evolution associated with the simulator demand may well differ from the forecast, although the two originate from the same decision rules. Now, rush and extra orders are triggered, if required. Specifically, if the demand cannot be entirely satisfied, the simulator triggers a rush order to fill this gap, whereas whenever the inventory of drug $f$ at the end of the day is below $\underline{s}^f$, a *restocking* order is triggered. Obviously, on the first day of the scheduling period, the demand forecast is likely to be close to the real demand, but this tie soon vanishes as time elapses. We expect pull orders to intensify at the end of each period. On this basis, in Section 6, we will experimentally test a modified pull order restocking policy that considers the time lag before the next push order for that drug, projects the current status in the ward for the next few days, assuming a consumption decreasing with time for that drug (thus realizing short-term forecasting), and orders the quantity required to cover the prediction in one shot. This policy aims to reduce orders of the same drug potentially triggered in consecutive days, and it is referred to as *aware* because it is an attempt to exploit fresh information coming from the actual status in the ward. In Section 6, we will experimentally test different values for $\omega$ and compare the aforementioned variants.

## 6 Computational results

### 6.1 Testbeds

As mentioned, we deal with the containment of the number of orders, given the constraints on budget (soft) and storage (hard). Recall that we distinguish regular orders, which have a one-day lead time and whose cost is the time nurses devote to them, from rush orders; these are a direct financial cost that affects hospital expenditure, are issued under pressure, and are served within two hours. Regular orders include both push orders, those pre-planned in the agenda, and the extra orders triggered by safety stock replenishment that can be issued at the end of the day. Order regularity is advocated because it helps nurses plan their activities.

Given the criteria to evaluate a solution, experiments aim to test the effectiveness of our strategy and all its variants. In particular, we aim to (*i*) assess the impact of the pre-planned agenda with respect to a classical inventory policy, which is described in Section 6.3, (*ii*) analyze the impact of the length $\omega$ of the scheduling period, (*iii*) compare the basic and the knapsack variants to set the safety stock level $\underline{s}^f$, (*iv*) assess the additional benefit of a demand-aware hybrid strategy introduced in Section 5.3, and (*v*) compare therapy-driven and drug-driven demand forecasting. We run a few sets of experiments, all considering one year as the time horizon $T^H$ and $\Omega = 28$ days as the duration of the planning period $T^\Omega$, whereas $\omega$ ranges in {7, 21} concerning the scheduling period duration. We run 30 repetitions of each experiment to smooth the effects of the specific demand realization.

### 6.2 Preliminary insights from a partial comparison with the real case

First of all, it has to be underlined that a full comparison between the as-is scenario and the results of the optimisation was not possible due to the mentioned critical issues about lack of data and difficulty in retrieving them. Specifically, as we mentioned in Section 4, we digitised information on orders placed and therapies administered for only one month (June 2016). The data collected cover 14 drugs, a subset of those used in the ward. In addition, for these drugs it was not possible to assume the level of stock in the warehouse at the beginning of the

**Table 10. Optimization results for varying capacity and budget parameters.**

| Capacity | Budget (€) | Number of orders |
| --- | --- | --- |
| 2 | 3.000 | 9 |
| 3 | 3.000 | 9 |
| 5 | 3.000 | 9 |
| 2 | 5.000 | 5 |
| 3 | 5.000 | 5 |
| 5 | 5.000 | 4 |

planning horizon. The optimisation model was therefore run assuming that only the drugs required on the first day were in stock. This assumption means that the number of orders placed by the model is potentially higher than that obtained from full knowledge of stock levels. For example, for a certain drug, it is observed that the first order in the as-is scenario occurred on day 7 while it was needed before. In the initial phase of the project, however, we performed a comparison between the as-is scenario and the optimized solution assuming a complete knowledge of the demand in the planning horizon considered (June 2016). We report in this section the results obtained at that time. The optimisation model used for this comparison is slightly different from the one described in this paper, which is the result of the project's progress. Specifically, the model of which we report the results in Table 10 (i) assumes a hard constraint on the budget (limited to 3000 or 5000 euros per day); (ii) does not consider the perspective of nurses; (iii) assumes a total capacity of 2, 3, or 5 shelves to store drugs (not organized in dedicated and shared storage spaces). The number of orders done in the as-is scenario is equal to 13, while in the optimized solutions the orders vary from 4 to 9.

Fig 5 reports some further information on the orders comparing the as-is scenario and the best optimal solution (4 orders) in terms of days in which orders are done ($x$-axis) and number of different drugs in an order ($y$-axis). Interestingly, the optimized orders are quite equally distributed in the planning period, and well balanced in terms of number of different drugs they

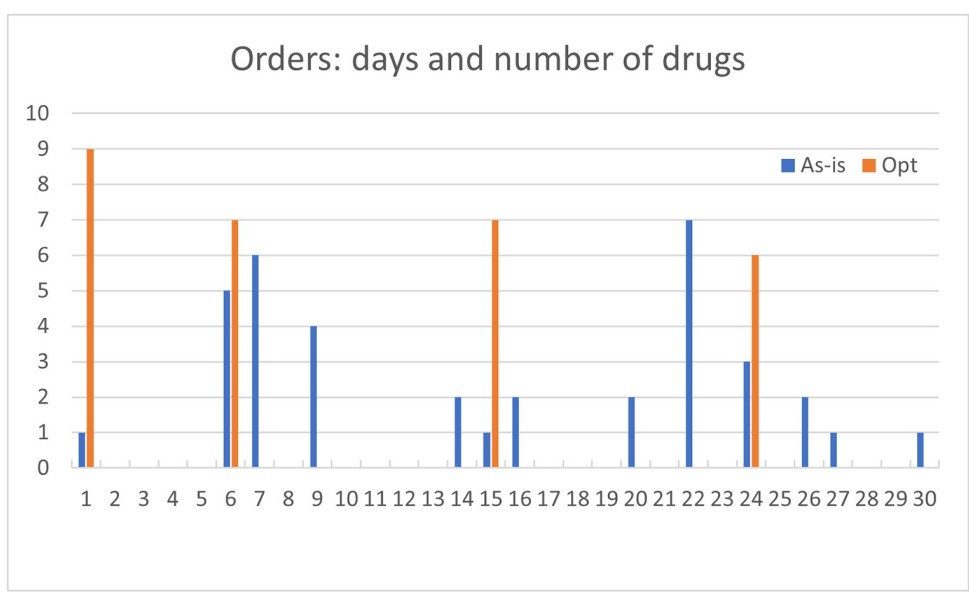

**Fig 5. Order information: The as-is scenario vs optimization (June 2016).**

contain. Clearly, the optimized solution suffers from the lack of information on the initial stock levels, as it is evident from the height of the bar on the first day. These results, even if preliminary, have been discussed with clinicians and have motivated further investigations.

We conclude by briefly commenting on the benefits of optimized drug order management. As noted earlier, rush orders have a much higher cost than standard orders. Reducing their number therefore has a significant impact on ICU costs. However, reducing rush orders is not the only benefit due to optimization. In fact, the reception of an order is a complex operation, composed by several tasks: check the consistency of what ordered with what arrived; check the consistency of what contained in the delivery note with what arrived; arrange the drugs in the warehouse; update the stock levels, only to quote the main ones. Receiving an order engages highly trained staff (nurses) in non-value-added activities. The time required by this activity is difficult to quantify since it depends on the degree of ward automation. Optimized drug order management allows this time to be used in patient care. Ultimately, an additional benefit of optimization is that having the drugs ready for use on the ward allows for early initiation of patient care. This is especially important on a ward where a delay in drug administration can be life-threatening.

### 6.3 A periodic review policy as a benchmark

To provide a benchmark for our strategy, we implemented a classical $(s, S)$ inventory policy based on periodic review. As we learned during our observation period in the ward and according to official national health service documents, replenishment is regularly performed by nurses once a week, usually on a Monday. The PAR level $S^f$ and reorder point $s^f$ for each drug $f$ have been estimated according to the literature.

As in [7], $s^f$ has been set as $s^f = d_f + \delta^f z$ according to [42], where $d_f$ is the average daily demand of drug $f$, $\delta^f$ is the standard deviation of the daily demand, and $z = 1.96$ is set according to the expected service level—97.5% with the demand normally distributed; the lead time of regular orders in our case is one day, so it does not appear in the formula.

Likewise, $S^f$ has been set as the sum of the reorder point $s^f$ and the *Economic Order Quantity* (EOQ), which is computed as follows (again, see [42]): $EOQ_f = \sqrt{(2D_f K)/(hc^f)}$ where $D_f$ is the annual average demand of drug $f$ in ward (in doses), $K$ is the estimation of the cost of receiving the order of one drug, considering a nurse hourly gross wage of 20€ and 6 minutes to handle the replenishment operations for each drug, thus yielding 2€, $h$ is the percentage of the value held in stock that is to be lost (cost opportunity), and we set it to 0.8, and $c^f$ is the cost of one dose of drug $f$. Note that such formulas are set in [42], taking for granted that demand is normally distributed. In our case, we can only exploit the empirical distribution we mentioned above to compute $D_f$ and $d_f$.

### 6.4 Experimental setting

In the following, we provide the details of the computational experiments concerning the simulation.

The simulation model was created using Rockwell Arena and Python. Each day, for each patient type, the simulation model creates a certain number of *patient entities*. The number of entities to create is determined by sampling from suitable empirical distributions based on the data collected on-site during the observation period. Upon arrival, each patient entity is assigned with a therapy. The therapy depends on the patient type and defines (*i*) the patient's LoS and (*ii*) the number of doses of each drug that he/she will be administered each day. The algorithm (coded in Python) that defines the therapy considers for each patient type the

conditional probability of a drug being needed on day $d$, given the drugs that were administered on day $d − 1$. Because of this, patients of the same type can be assigned with different therapies. Once a patient entity is assigned with a therapy, the patient seizes a ward bed. The bed will be subsequently released after a time interval equal to the LoS.

Every day, a *nurse entity* is created. This nurse checks the drugs needed by each patient in the ward (which depends on the therapy) and administers the drugs, if available. If the amount of drug in stock is smaller than the one required, the nurse issues a rush order. Upon drug administration, the stock level for each administered drug decreases. The stock level is updated after the rush order lead time. Once the drug is restocked, the pending administration eventually takes place. Then, for each administered drug, the safety stock is checked and an extra order is issued, whose quantity will either restore $\underline{s}^f$ or a larger quantity if the *aware* option is active.

Every $\omega$ days (e.g., 7 or 21), the simulation model creates an *order entity* that triggers an algorithm (either an optimization model or an ($s$, $S$) heuristic, depending on the scenario) that indicates for each drug the number of boxes to order. The algorithm is executed in shell (i.e., while the algorithm runs, the simulation clock does not advance), and once a solution has been computed, the simulation model reads it and triggers pre-planned orders accordingly. After the regular order lead time has elapsed, the model updates the stock levels by adding the ordered quantity.

In this study, we have considered 18 drugs, 6 patient types, a ward capacity of 8 beds, a lead time for regular orders equal to 1 day, and a lead time for rush orders equal to 2 hours. A rush order costs 300€ for each drug involved in the order. The real cost of the 18 drugs has been used: it ranges from 0.30€ to about 113€ per dose. Drug boxes may contain from 1 to 25 units, and the volume of the smallest item is 100-fold of the largest volume, whereas the daily dosage for a drug ranges from 1 to 8.

## 6.5 Variants of the hybrid inventory policy: Numerical results

We report and discuss the results of the computational experiments deployed to answer research questions (*i*) to (*iv*) set in Section 6.1.

First, we focus on the number of rush orders (if a rush order involves two different drugs, it is counted as 2), which is proportional to the direct cost associated with this practice. We report the boxplots of the results of the 30 instances for each variant in Fig 6, assuming that the generator is used to forecast demand. Specifically, for $\omega = 7$ and $\omega = 21$, we report all four combinations concerning the safety stock strategy (basic vs. knapsack) and reorder policy (aware vs. not aware). The last boxplot refers to the ($s$, $S$) policy benchmark.

At a glance, one can see that the ($s$, $S$) rush order cost can be halved or even decreased to 1/18. In particular, if we compare each basic variant with its knapsack version, we see that in each version, the latter always dominates the former. In general, this can be expected because the safety stock is intended to hedge against demand fluctuation avoiding rush orders; it also means that the drugs whose stock has been increased have been wisely chosen. The savings is larger when the basic policy cost is higher, which happens for the not aware basic variants. If we compare the impact that the change from basic to knapsack provides in case of aware policies, this is still present but not so remarkable. This means that the rush order containment that can be granted by a higher level of stock always present (knapsack policy) can almost entirely be achieved by a wise (aware policy) replenishment sizing. Indeed, the aware policy adapts the replenishment lot size to the patients currently in the ward and to the time lag to the next push order for that drug in the agenda, thus avoiding the storage of large amounts of drugs when the demand is low. Finally, concerning $\omega$, we have an unexpected finding: despite

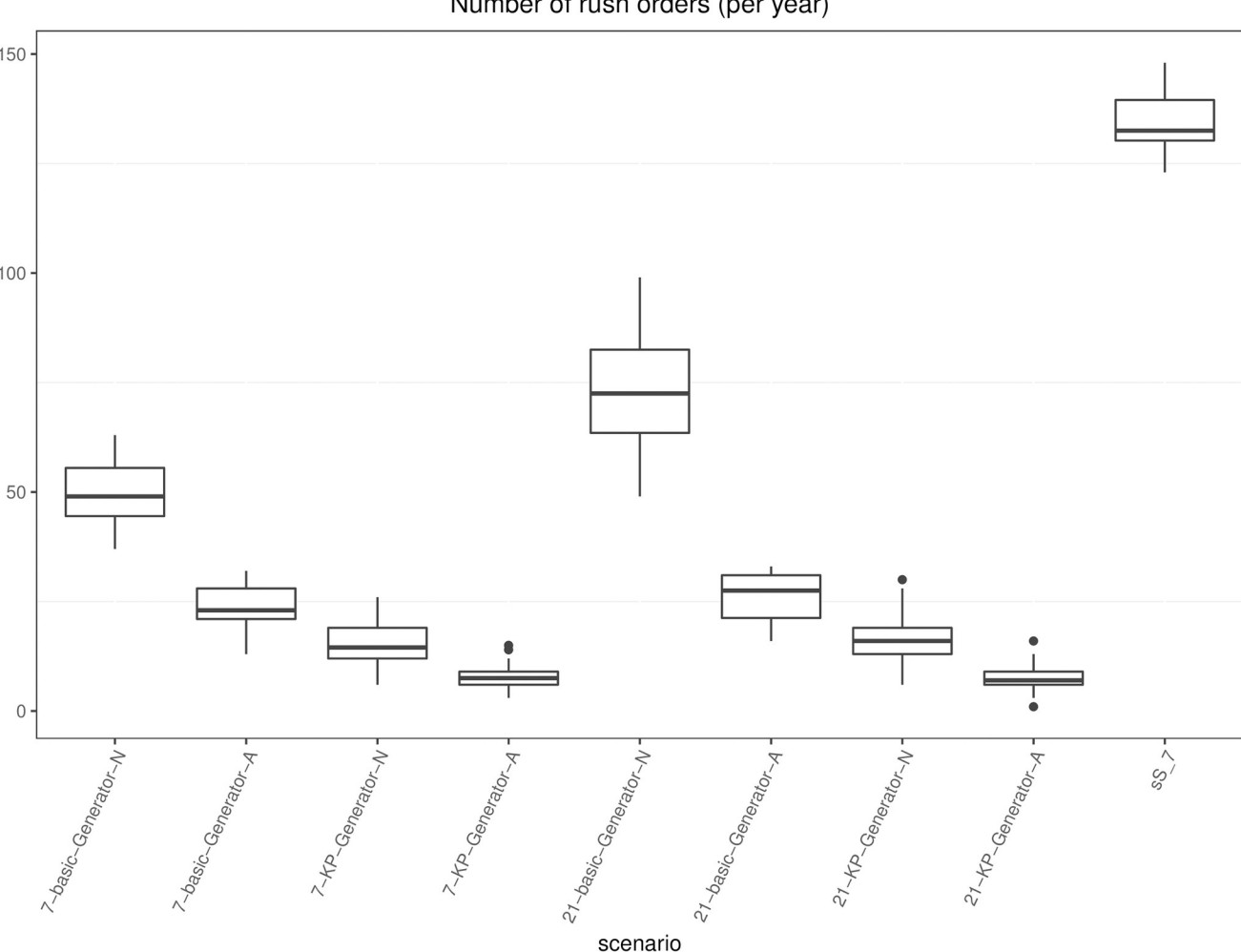

**Fig 6. Boxplots of the number of rush orders.** Each boxplot shows the results of 30 instances. Policy names refer to the $\omega$, safety stock option (basic or knapsack), demand forecasting option (Generator), and aware (A) or not aware (N) variant.

having increased the width of the scheduling period $\omega$ (from 7 to 21) and, therefore, the potential gap from the forecast to the realized demand, the number of rush orders remains almost stable which represents a noticeable result. As a whole, the most effective policies are the knapsack aware ones, with 21 certainly preferred to 7 by nurses because of the longer period agenda.

Note that the best-performing policies lead to a maximum number of rush orders per year less than 20 (namely, 15 for 7-KP-Generator-A and 16 for 21-KP-Generator-A), compared with a maximum of 148 rush orders for $(s, S)$. On average, the number of rush orders for $(s, S)$ is $134.6\bar{6}$, whereas the figure is 7.3 for the best-performing policy. Given a cost of 300€ for each rush order, the saving is about 38.000 €. This more than compensates for the larger budget devoted to inventory required by our policies; see Fig 7. There, the average daily value of the drugs tied in stock is depicted. Policy $(s, S)$ holds about half the inventory value with respect to ours. However, as opposed to manufacturing, in health care, stocked drugs are not such a direct cost, and they provide a safety margin to sudden therapy changes. In addition, we observe that the maximum increase of daily stock value between our policies and the $(s, S)$

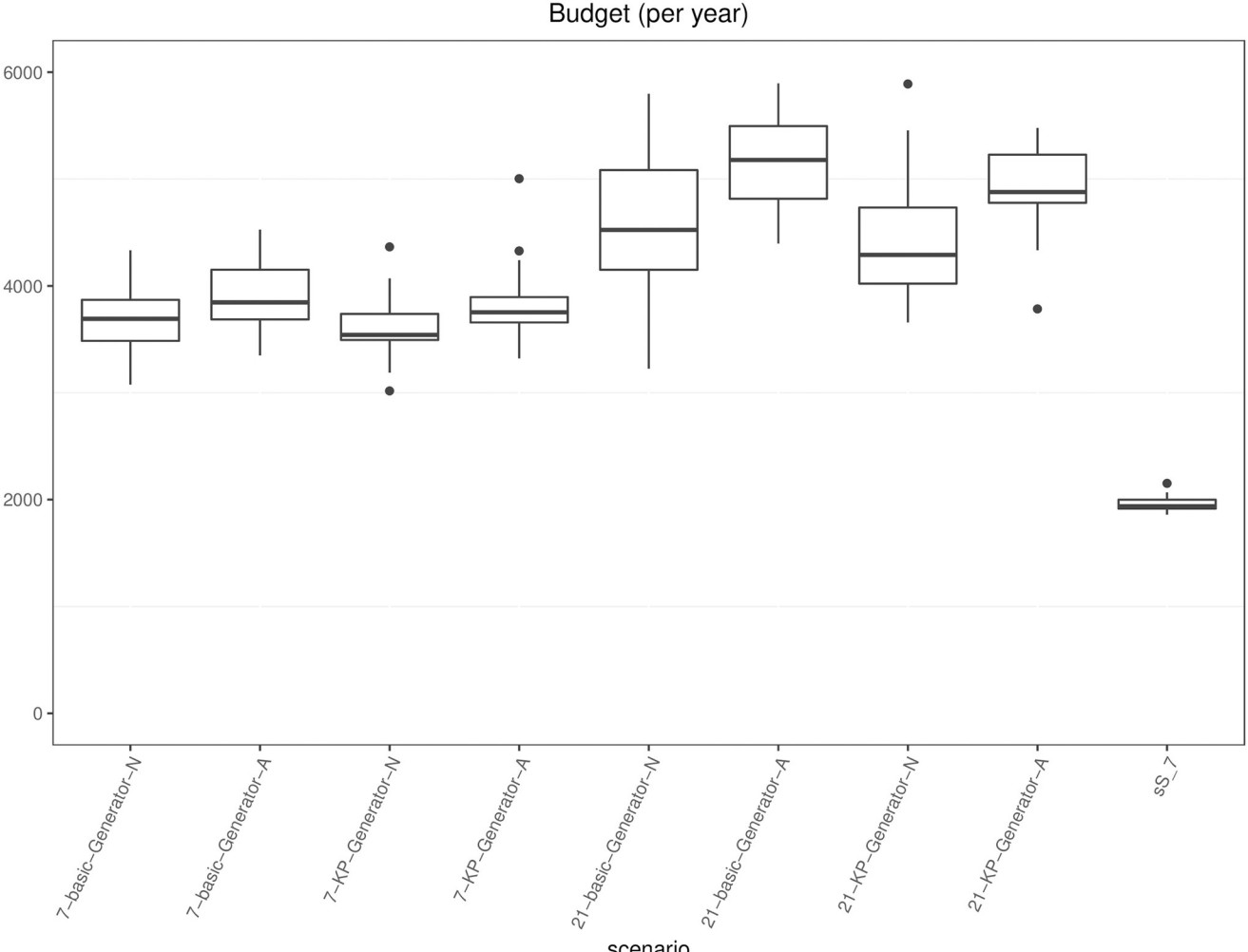

**Fig 7. Stock value.** For each variant, the average value of drugs in stock on a day, over the year (€). Each boxplot shows the results of 30 instances. Policy names refer to the $\omega$, safety stock option (basic or knapsack), demand forecasting option (Generator), and aware (A) or not aware (N) variant. sS_7 refers to the periodic policy.

policy is not over 3.000€, and if we assume an inventory cost roughly equal to 1% of the total stock value, it would yearly affect the hospital expenditure for about 10.000€, which amounts to about one third of what our policies save in rush order cost. Note that as $\omega$ increases, both the average and the variance of the average daily inventory value tend to increase. Indeed, as the scheduling period becomes larger, it is more likely that the drugs present in the forecast scenario are not used in practice and remain in stock for a while, increasing the inventory value.

Finally, let us turn to the number of orders and the number of different drugs involved in each order, on average. The number of days on which an order event occurs is depicted in Fig 8, separately for push, replenishment, and total orders. Concerning push orders, depicted on the left hand side of the picture, their number decreases as the scheduling period $\omega$ increases, as expected. Basic aware seems to lead to more ordering days rather than basic not aware. Because awareness affects the lot size of pull orders, the effect on push orders is undirected by way of inventory levels at the beginning of each planning period. As just noticed, aware

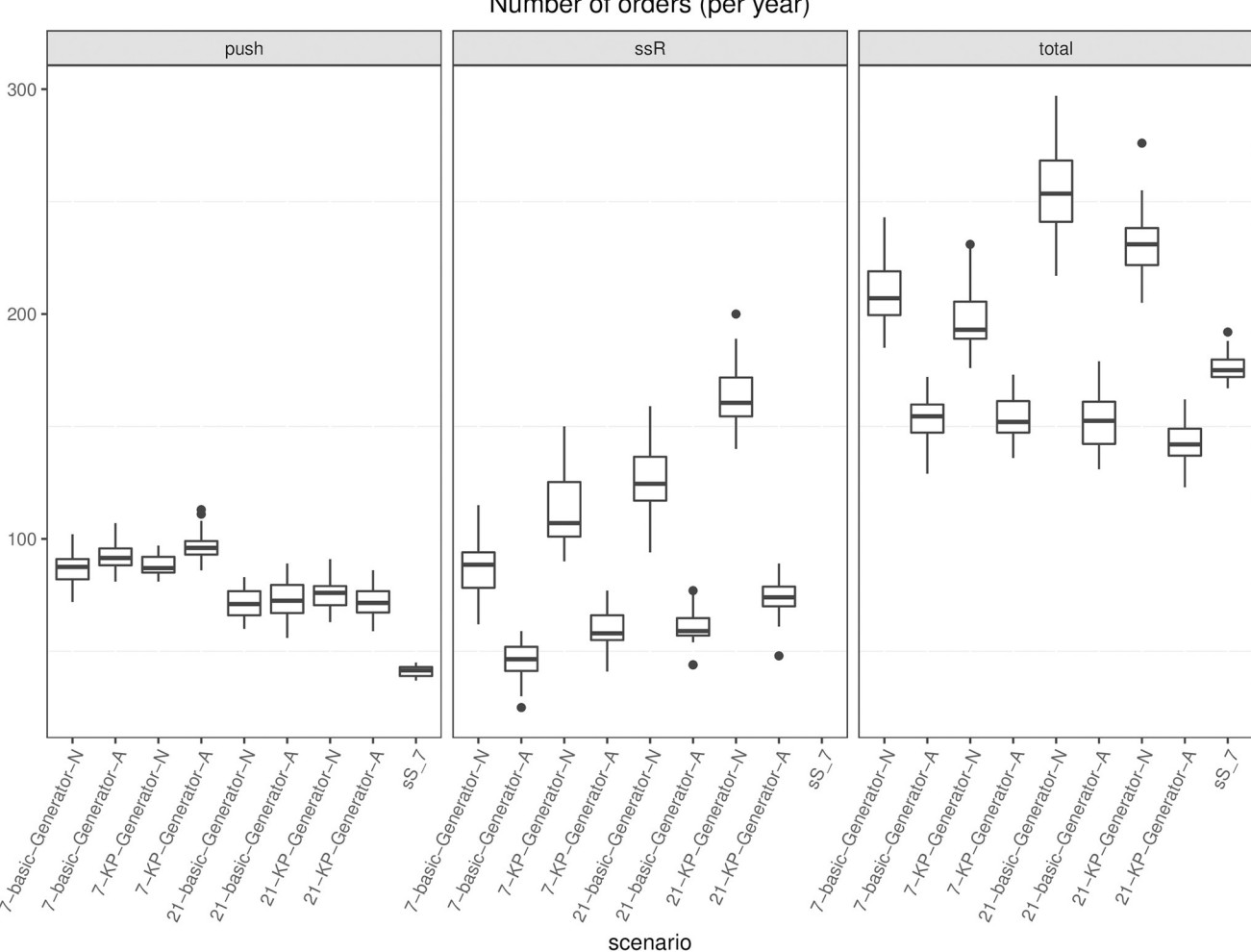

**Fig 8. Boxplots of the orders (push, replenishment, all).** On the left are the boxplots of the number of days with push orders per year. In the middle is the number of days with replenishment orders per year. On the right part, all orders are considered, push and pull. Each boxplot refers to 30 instances.

reordering may allow a lower average level of stock, consistent with the basic policy. A low inventory may then raise the number of ordering days in the agenda. On the other hand, the knapsack policy tends to keep a large inventory, so the awareness effect is negligible. We considered as push orders the periodic ones in the $(s, S)$ policy because they are scheduled a priori. Their number is almost constant because of periodicity. Recall that pre-scheduled ordering is preferred by nurses with respect to just-in-time ordering.

The middle part of the picture clearly shows that unaware policies entail a greater number of replenishment orders compared with the aware ones, and the number of replenishment orders is more stable in the aware variants. On the right hand side of the picture, the number of ordering days per year is reported, pull and push orders together. Interestingly, we see that all the aware variants exhibit a lower number of total orders than $(s, S)$, thus confirming the importance of exploiting the information coming from the drug management process.

Now, we look at the average number of drugs in an order, as depicted in Fig 9, regarding push orders (on the left column), pull orders for safety stock replenishment (in the middle), and with respect to all orders (on the right). Policy $(s, S)$ scores the lowest number, but the variance, -as far as the boxplot conveys this information, is comparable to that of the other

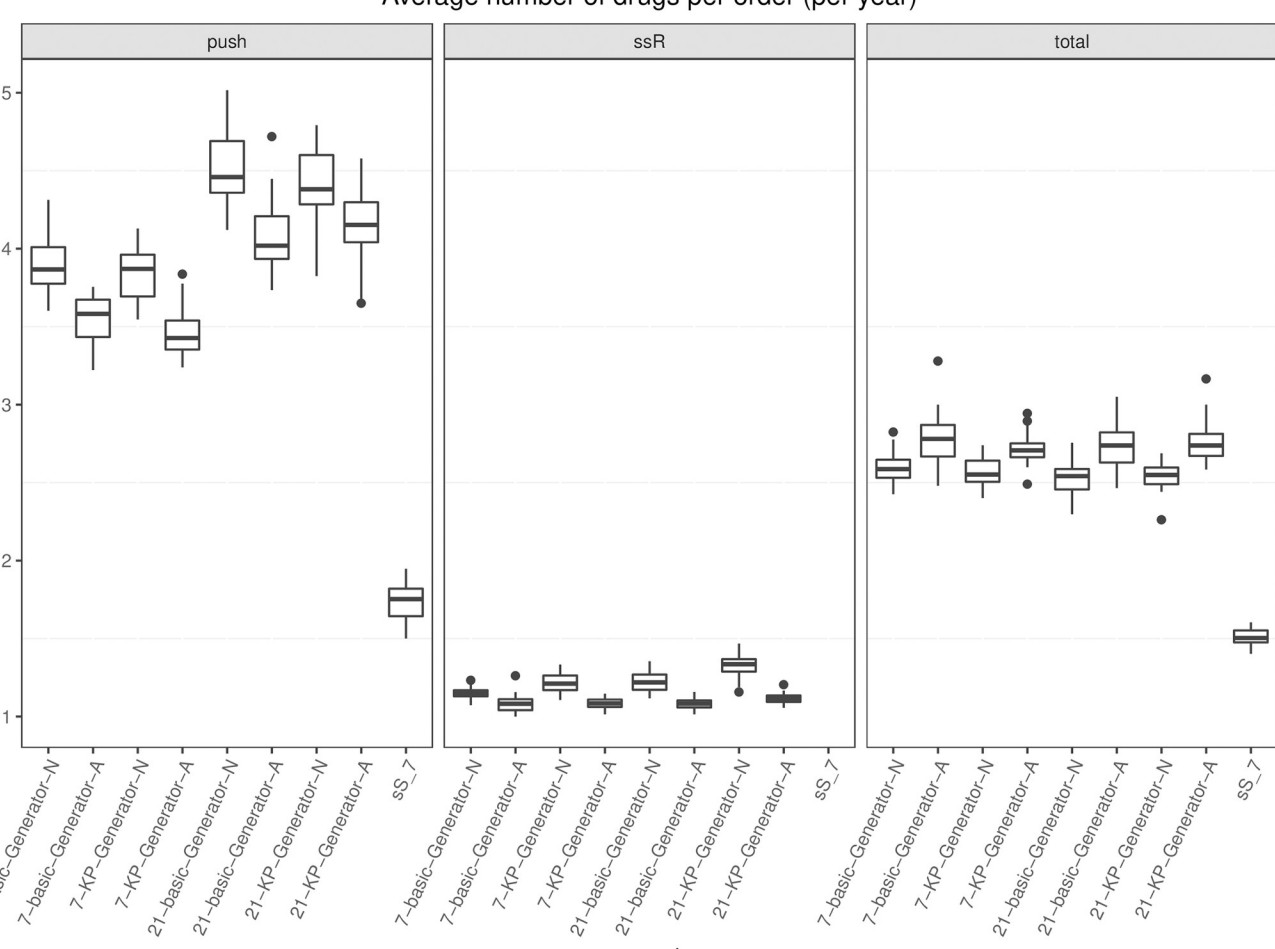

**Fig 9. Boxplots of number of drugs (push, replenishment, all).** On the left are the boxplots of the average number of drugs in push orders only. In the middle, only safety stock replenishment pull orders are considered, and the average number of drugs in all orders is shown on the right. Each boxplot refers to 30 instances.

policies. Awareness seems to reduce the number of drugs in safety stock replenishment (see *ssR* in Fig 9), probably because of the fact that different drugs may have quite different inventory levels and fall below the threshold on different days. The $(s, S)$ lowest total average may be related to the frequent rush orders that typically involve single drugs.

In conclusion, the results suggest that we can pre-schedule on a long horizon without deteriorating performance, and the knapsack policy and aware restocking are essential tools to contain the number of stock outs. Indeed, the former can be seen as a priori guess on the future demand, while awareness exploits current information (patients' status) and the knowledge of the agenda to tailor restocking orders a posteriori.

## 6.6 Therapy-driven versus drug-driven forecasts: Numerical results

We report and discuss the results of the computational experiments used to answer research question ($v$) set in Section 6.1. In this section, we compare the results given by the two different types of forecasting methods, which are therapy driven and drug driven. As described in Section 4.3, the first method (therapy driven) estimates the demand of drugs based on ($i$) the

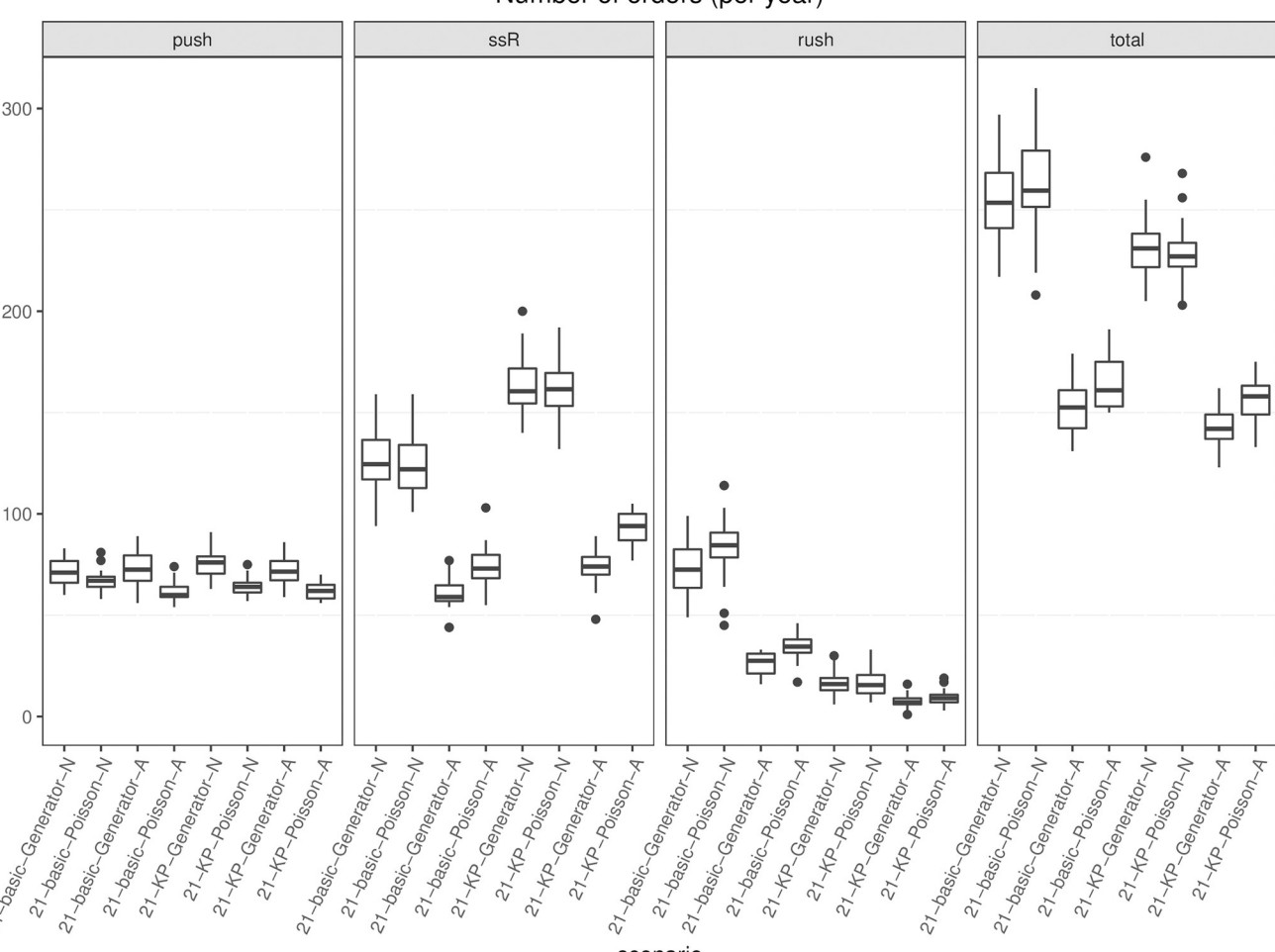

**Fig 10. Boxplots of the orders (push, replenishment, all).** On the left are the boxplots of the number of days with push orders per year. In the middle are the replenishment and rush orders. On the right part, all orders are considered, push and pull. Each boxplot refers to 30 instances.

current number of patients in the ward, (*ii*) the therapy that each patient is currently undergoing, and (*iii*) the possible evolutions that may occur in each therapy because of changes in the patient's condition.

The second method (drug driven), by contrast, is based on a simpler Monte Carlo simulator, that generates, for each drug and for each day, a discrete demand expressed in number of doses. Such a demand is obtained by sampling from Poisson distributions (one for each drug *f*) with parameter $\lambda_f$. The parameter $\lambda_f$ represents the expected value of the daily demand, expressed in doses for the drug *f*. The idea of using Poisson distributions in the Monte Carlo simulation is taken from the literature [27].

In this section, we report the results for the longest scheduling period only ($\omega = 21$); the results for the variants with $\omega = 7$ are similar. Specifically, we show in Fig 10 the boxplots relative to the number of orders for the eight variants resulting from considering $\omega = 21$ combined with each of the two safety stock options (basic or knapsack), with each of the two demand forecasting options (Generator or Poisson), and awareness or not. We observe that for the given safety stock option and awareness option, the numbers of total and rush orders obtained when the demand prediction is therapy driven are smaller than the corresponding values

Average number of drugs per order (per year)

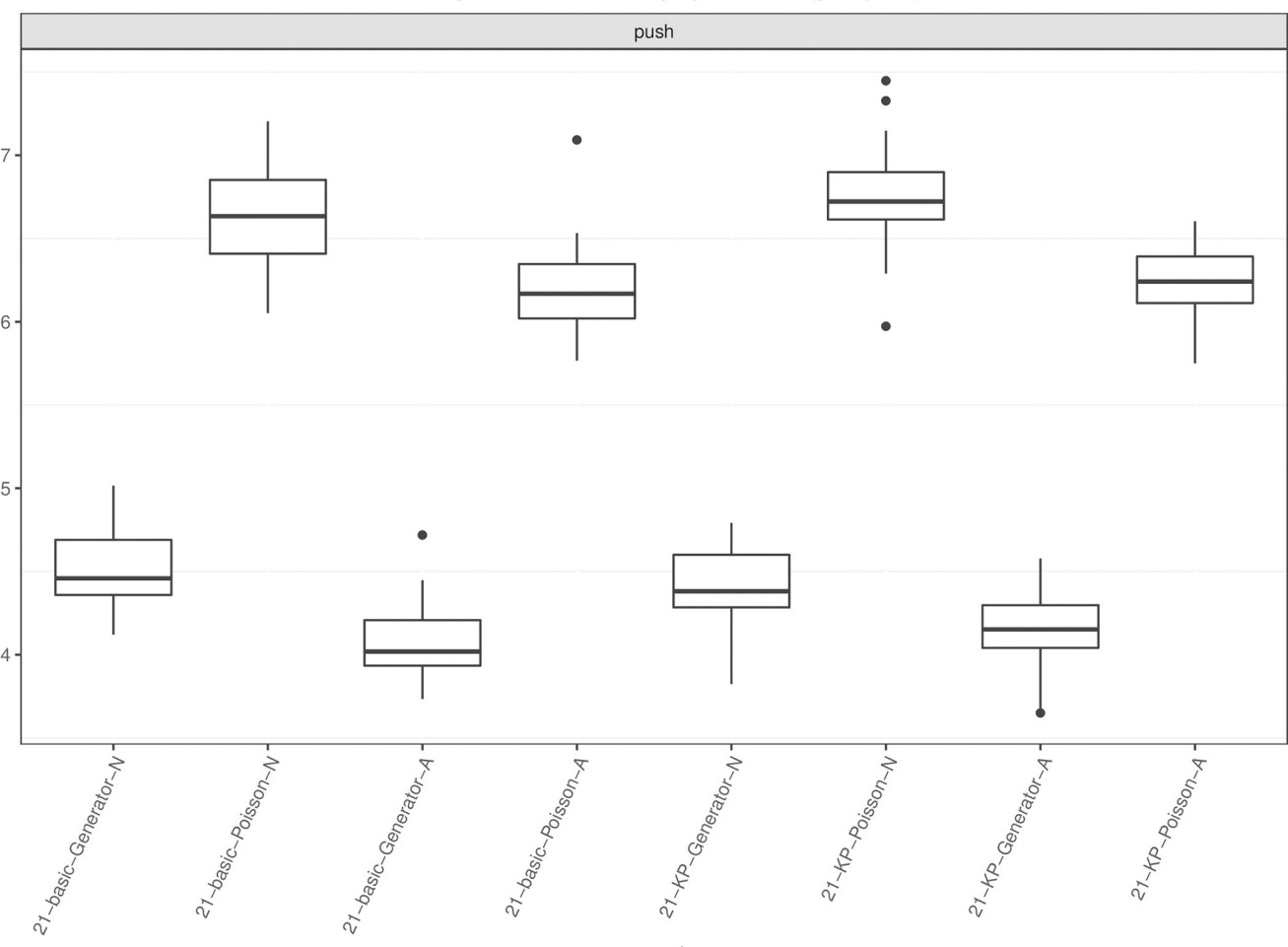

**Fig 11. Boxplots of the average number of drugs in push orders.** Each boxplot refers to 30 instances.

obtained with the drug-driven prediction. The difference between the two policies tends to decrease as we progressively incorporate knowledge within them. The drug-driven variants tend to give a smaller number of push orders, but those orders are characterized by a quite remarkable higher number of drugs per order, as Fig 11 clearly shows. On the contrary, with the drug-driven forecast, the budget is, on average, 21.4% smaller than that obtained with the therapy-driven forecast.

In summary, the analysis reveals that therapy-driven policies are more accurate than drug-driven ones and lead to better results in terms of rush orders whose containment is our first goal.

## 7 The decision support system: Prototype of the graphical user interface

In the data collecting phase of the project, we created a desktop application with the twofold objective of (i) allowing a rapid transcription of the patient therapies; (ii) providing a tool to help clinicians and nursing staff to enter the data automatically. Specifically, we chose a desktop application rather than a web application to speed up implementation and reduce

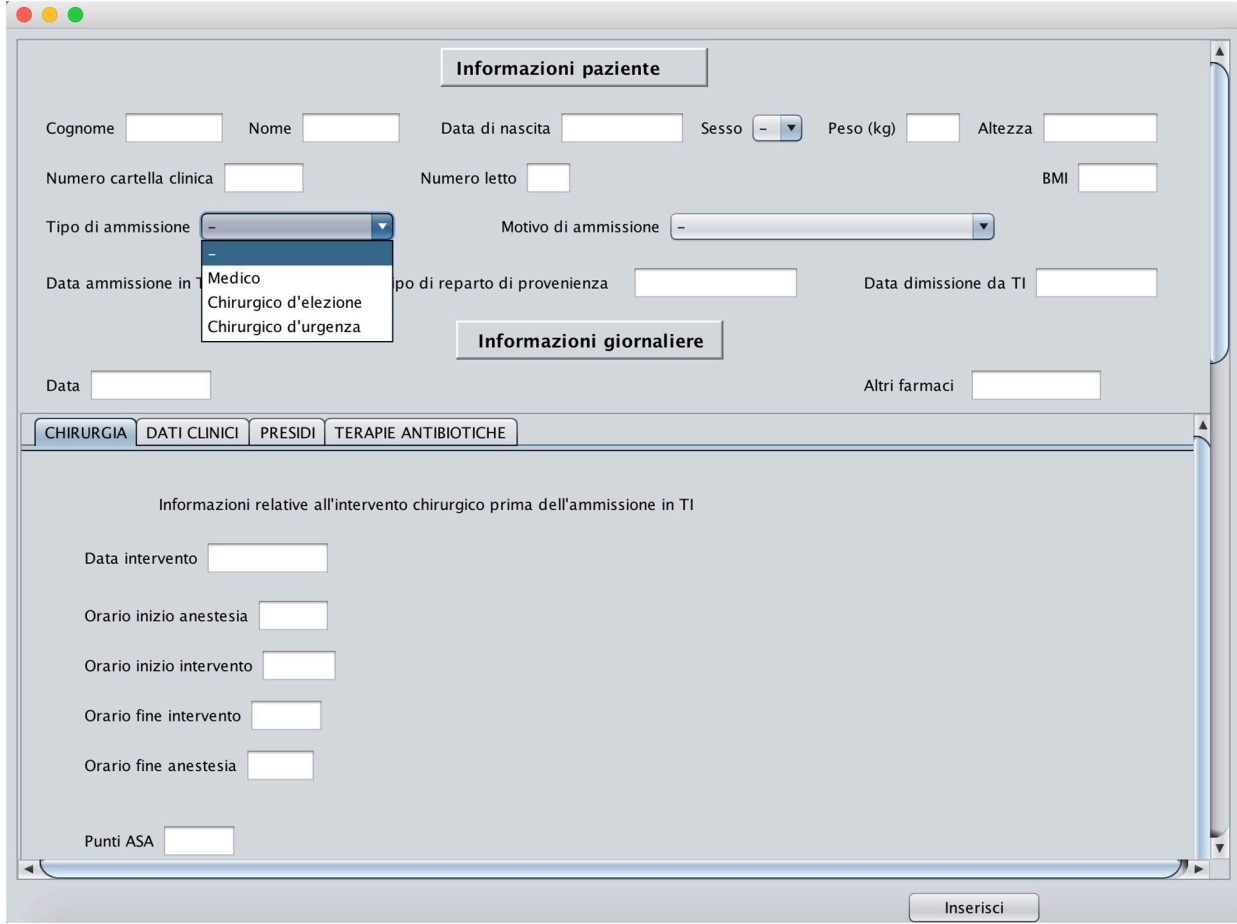

**Fig 12. General information on patients and panels for specific information.**

bureaucracy. The application uses the Swing Java library and follows a top-down event-driven approach: the main graphical area contains general information on a certain patient and is organized in panels containing specific information that are filled when some event occurs. As an example, Fig 12 is a screenshot of the main graphical area, while Figs 13 and 14 show the panels to fill respectively when results from the lab arrive and when clinicians set the target therapy. Text is in Italian as required by hospital's management.

This simple application was designed as a building block for a future decision support system (DSS) organized in a patient section and a ward section. The objectives of such a system are: (i) collect and make easily accessible patient-related information from different sources; (ii) provide a snapshot of patients admitted and resources allocated at a given time; (iii) allow automatic drug order management; (iv) display the prediction results about LoS and infection outbreak to support clinicians in making decisions and to allow a better management of the department; (v) facilitate the exchange of information with other departments: for example, if a patient was discharged from the ICU to be admitted to another department, information about the patient (treatments, evolution of clinical conditions, etc) could be shared. As another example, information (and prediction) about future bed availability could be used by a surgical department to plan surgeries.

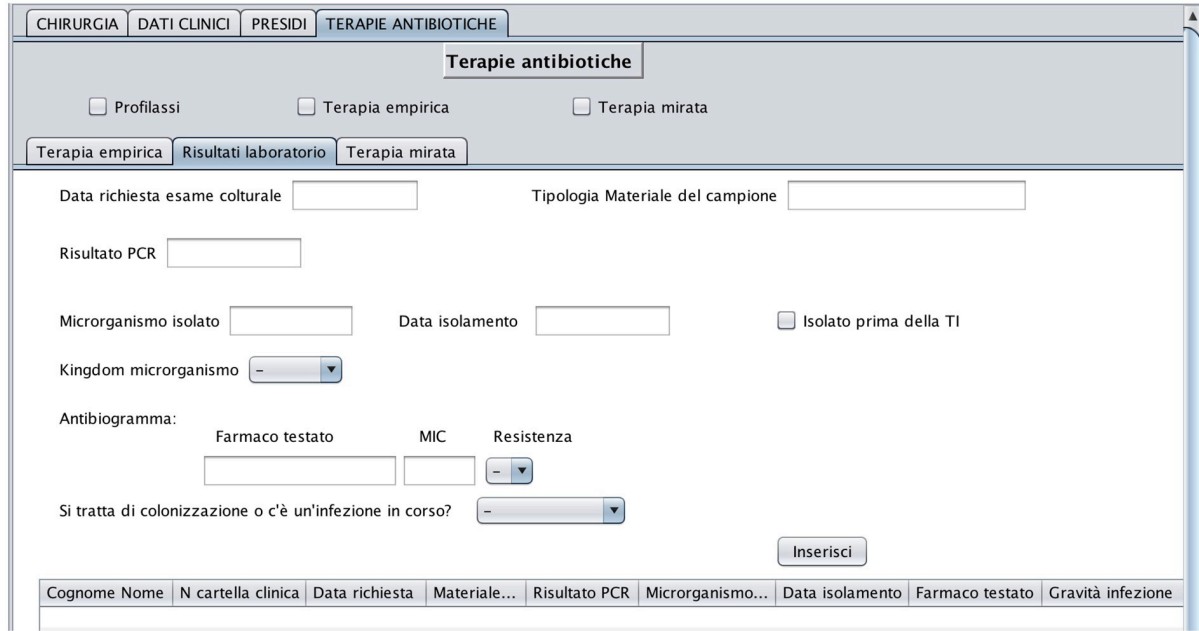

**Fig 13. Therapy panel (Terapie antibiotiche) and lab sub-panel (Risultati laboratorio).**

Ideally, the main decision flows concerning drug replenishment that would occur if our tool were deployed, are depicted in Fig 15. Clearly, demand simulation is no longer needed to mimic the realization of the real demand in the ward. Demand prediction on each forthcoming planning period is used to feed the optimization module on the first day of each scheduling period.

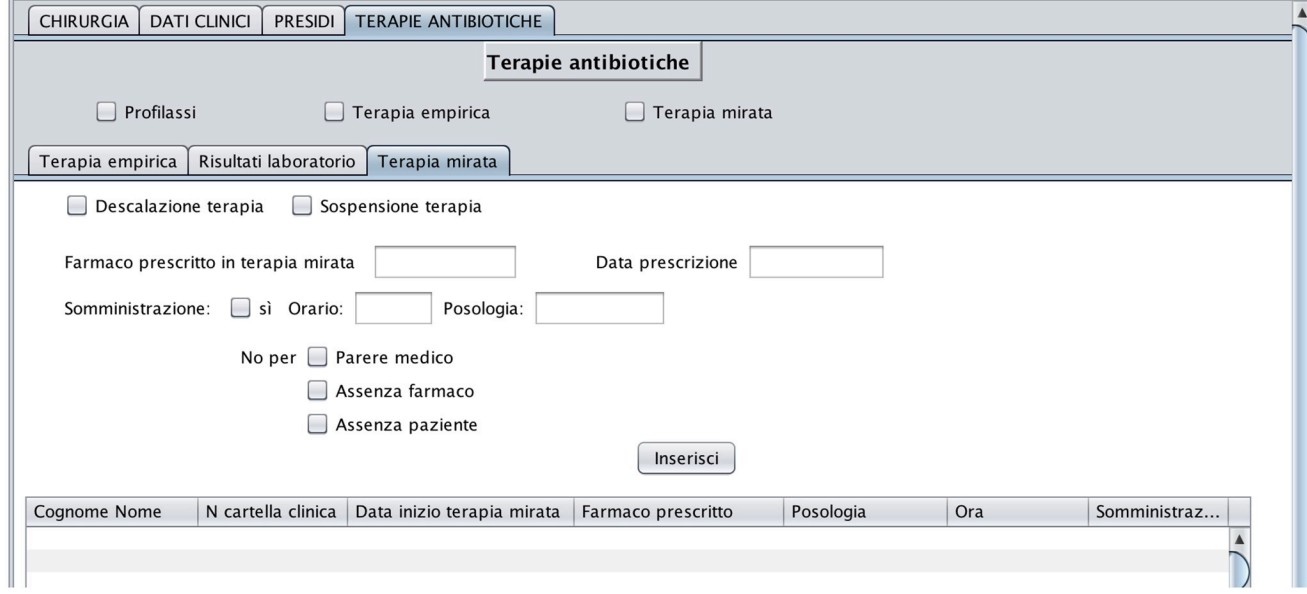

**Fig 14. Therapy panel (Terapie antibiotiche) and target therapy sub-panel (Terapia mirata).**

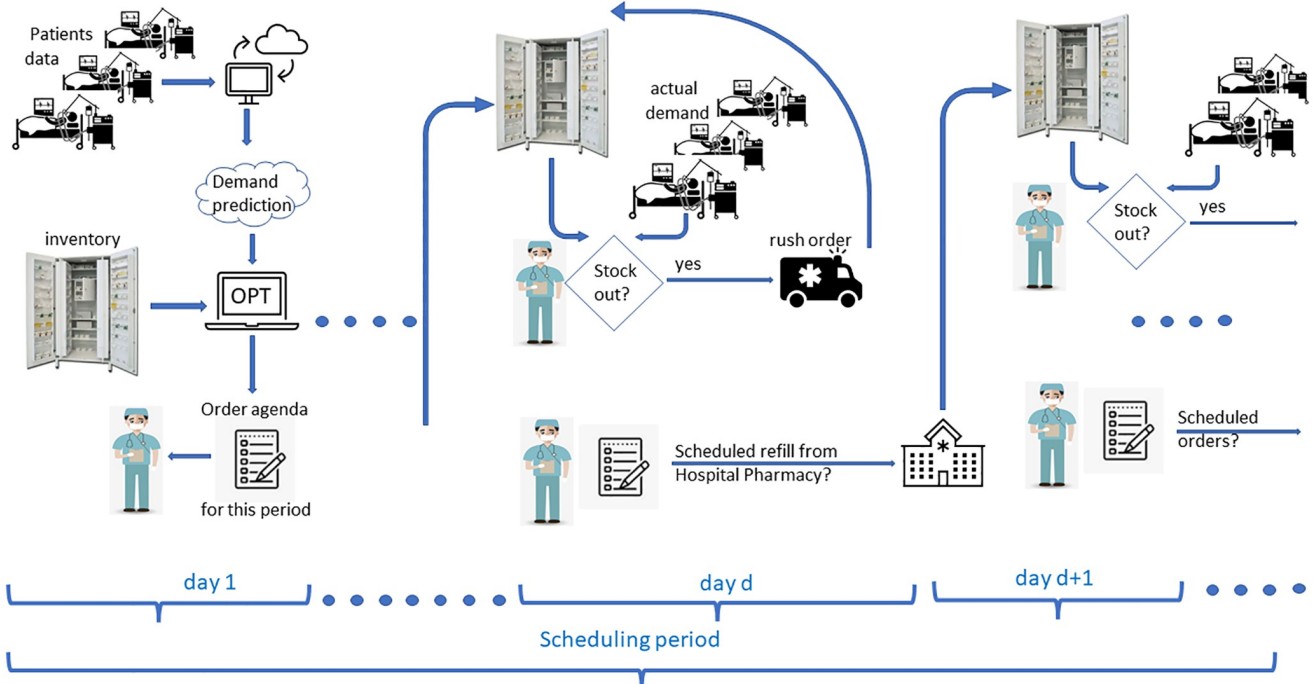

**Fig 15. Main decision flows in the deployment of the tool.** Prediction and optimization modules are launched at the beginning of each scheduling period.

The DSS briefly discussed in this section is intended to be only a first attempt of providing a tool tailored to the specific needs of a ICU.

## 8 Conclusions and work in progress

We proposed and tested a replenishment policy for critical drug management in ICUs, whose demand cannot be estimated by traditional means. Demand forecast is based on knowledge of therapies and the current patient population in the ward. Specifically, we proposed a hybrid push-pull inventory policy and evaluate its performance when used in a rolling horizon framework. Push orders are scheduled by solving an MILP model receiving in input the forecast demand. The optimization model minimizes the number of order occurrences (number of days in the period when an order is issued), which is considered a KPI in this setting. Another crucial target of the optimization model is to guarantee regularity in order management, which is another important KPI. Regularity concerns the order size, and the model imposes that the same size for each order of the same drug always occurs within each planning period. Pull orders come into play when the realized demand deviates from the forecast. Several variants of the hybrid push-pull inventory policy have been proposed and experimentally investigated by varying the length of the horizon in which the solution provided by the optimization model is considered reliable (7 and 21 days). The longer the horizon is, the higher the confidence in the optimization model is and, consequently, the smaller the frequency with which the optimization model is run and the higher the regularity is. In addition, we consider two alternative ways to fix safety stock, namely, *basic* and *knapsack*. For each combination of type of safety stock and frequency of optimization runs, we also investigated an *aware* variant of the push-pull policy that attempts to exploit information from the current status of the ward. Indeed, in the not aware variants of the policy, both stock out and safety stock deficiencies

trigger a pull order, either rush or regular. In this case, the ward asks for the minimal quantity needed to satisfy demand or restore safety stock. In the worst case, however, the unpredicted arrival of a patient requiring a particular drug not in stock triggers a sequence of periodic pull orders, according to the therapy. An expert nurse would recognize the case and thus increase the order size to a greater level than the safety stock if the next pre-planned order of the drug lies further in time. The aware variants of the policy mirror the strategy of the expert nurse, thus exploiting this kind of information. To further corroborate the importance of incorporating information in the inventory policy, we also evaluate the impact of using a therapy-driven forecasting tool as opposed to a drug-driven tool.

The results stimulate a very interesting discussion. First, they allow the conclusion that each of the policies proposed significantly outperforms the standard periodic review policy used as a benchmark in terms of the number of stock outs. This is particularly important because stock outs represent a prohibitive cost. In addition, they are a source of inefficiency and clinical risk within a hospital. Second, the use of information serves as a crucial tool through which the performance of the push-pull policy is improved. Third, the policy proposed seems to be quite robust with respect to the length of the time horizon, thus allowing the conclusion that trusting the optimization model for a longer period does not deteriorate performance and that indeed, it allows an increase in regularity. The proposed models allow to simplify the work of nurses working in ICUs, and to reduce the waste of the (expensive) drugs they use. Consequently, they contribute to optimize the operation of organizational units (the ICUs) which is of paramount importance especially during a pandemic such as the one related to SARS-CoV-02. However, the proposed push-pull policy does not come without limitations: drugs can indeed be ordered to satisfy the forecast demand and remain in stock if the realized demand deviates from the forecast one. Drugs in stock thus consume budget and affect the next orders. Even if, as mentioned, inventory costs are not a direct cost in wards, a better use of the budget should be done. This can be accomplished in at least two future research directions: (*i*) allowing a dynamic management of the budget along the planning horizon and (*ii*) experimenting collaborative policies between two or more wards according to which they exchange drugs to cope with demand variability. Preliminary results on the effect of lateral transshipment between collaborating wards are shown in [43]. Finally, an interesting research line concerns the introduction of uncertainty in the optimizer. Specifically, at the moment being, we assume for each patient that drug consumption is given by one of the paths representing the possible evolutions of his/her clinical condition, as provided by the demand generator. Indeed, the forecast path can be very different from the realized one in terms of resource (drugs and bed) utilization, thus vanishing the potential benefits coming from optimization. Taking into account demand uncertainty surely deserves further investigation.

## Acknowledgments

The authors of this paper would like to express their sincere thanks to Prof. De Gaudio and all his staff who supported us in all the phases of the project. They would also like to truly thank Dr. Villani from Cremona Hospital (Italy) for the stimulating and supporting discussions we had during the revision of this work, and Dr. Bagattini for his advice on prediction tools.

## Author Contributions

**Conceptualization:** Paola Cappanera, Maddalena Nonato, Filippo Visintin.

**Data curation:** Roberta Rossi.

**Formal analysis:** Paola Cappanera, Maddalena Nonato.

**Funding acquisition:** Paola Cappanera, Filippo Visintin.

**Investigation:** Paola Cappanera, Maddalena Nonato, Filippo Visintin.

**Methodology:** Paola Cappanera, Maddalena Nonato, Filippo Visintin, Roberta Rossi.

**Project administration:** Filippo Visintin.

**Software:** Paola Cappanera, Maddalena Nonato, Filippo Visintin, Roberta Rossi.

**Supervision:** Paola Cappanera, Maddalena Nonato, Filippo Visintin.

**Validation:** Paola Cappanera, Maddalena Nonato, Filippo Visintin, Roberta Rossi.

**Visualization:** Roberta Rossi.

**Writing – original draft:** Paola Cappanera, Maddalena Nonato, Filippo Visintin, Roberta Rossi.

**Writing – review & editing:** Paola Cappanera, Maddalena Nonato, Filippo Visintin.

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
