## [Decision Letter · Decision Letter 0]

1 Jul 2021

PONE-D-21-11825

Rush order containment of critical drugs in ICUs

PLOS ONE

Dear Dr. Cappanera,

Thank you for submitting your manuscript to PLOS ONE. After careful consideration, we feel that it has merit but does not fully meet PLOS ONE’s publication criteria as it currently stands. Therefore, we invite you to submit a revised version of the manuscript that addresses the points raised during the review process.

We look forward to receiving your revised manuscript.

Kind regards,

Behzad Behdani

Academic Editor

PLOS ONE

Journal Requirements:

 [The work by P. Cappanera, R. Rossi and F. Visintin has been partially supported by the LINFA (Logistica INtelligente del FArmaco) project, funded by Regione Toscana under the call PAR FAS 2007-2013, Linea d'azione 1.1 - Bando FAR FAS 2014.

https://www.regione.toscana.it/-/bando-far-fas-finanziamento-di-progetti-di-ricerca-nei-settori-energia-fotonica-ict-e-robotica

The funders had no role in study design, data collection and analysis, decision to publish, or preparation of the manuscript.]. 

Reviewers' comments:

Reviewer's Responses to Questions

**Comments to the Author**

1. Is the manuscript technically sound, and do the data support the conclusions?

Reviewer #1: Partly

Reviewer #2: Yes

2. Has the statistical analysis been performed appropriately and rigorously? 

Reviewer #1: Yes

Reviewer #2: Yes

3. Have the authors made all data underlying the findings in their manuscript fully available?

Reviewer #1: No

Reviewer #2: Yes

4. Is the manuscript presented in an intelligible fashion and written in standard English?

Reviewer #1: Yes

Reviewer #2: Yes

5. Review Comments to the Author

Reviewer #1: Summary

The authors present a decision support system to manage the inventory policy of medications with low and intermittent demand at the ICU setting. They propose a simulation approach to forecast the demand and create scenarios of usage within the hospital system. They solve the inventory problem using a MILP model based on the principles of thee Lot-sizing problem. The study focuses on demand estimation on a rolling horizon, incorporating the properties and medical guidelines related to drug consumption. Thus, the tool is driven by patterns of therapies rather than patterns of drugs. The computational experiments investigate the effect of the model parameters and variants and provide a direct comparison with a standard inventory policy frequently encountered in practice.

The paper proposes a flexible quantitative methodology to solve an important challenge faced by decision makers in practice. The presented analysis highlights the benefit of this approach and its potential to positively affect the efficiency of major healthcare systems. Below you will find my observations regarding specific issues that need to be addressed to improve the paper.

Major Comments

1. The paper would be greatly improved if it included a direct quantitative and qualitative comparison of the proposed solution with what has been currently implemented in practice. My biggest concern relates to the accuracy of the forecasting tool and thus I strongly suggest a one-to-one evaluation of the inventory planning framework with the results obtained at the hospital in the same period. In other words, the authors could run a hypothetical retrospective “trial” in which they use for some period of time the proposed system in parallel to what is currently implemented in the clinic. This analysis would highlight the true benefit and applicability of the decision support tool.

2. A suggestion would be to add a section regarding the deployment and implementation of the tool in practice. What are the challenges that need to be overcome for its integration? How would the interface for the decision maker look like? This addition in combination with the one-to-one comparison will significantly improve the manuscript.

3. I would recommend to modify the abstract and emphasizing more the key findings and conclusions of the investigation. The current structure is not very coherent.

4. Why does the forecasting tool not take into consideration the patients that are scheduled to be admitted to the hospital in the planning period in addition to those currently hospitalized? Scheduled surgeries for example depending on their severity are associated with the risk of ICU admission. Thus, they could be factored as part of the forecasting tool.

5. The section regarding the demand generation process and forecast needs to be substantially refined and expanded. The authors should discuss how they learn the probabilities of the proposed flowchart and in which ways the patient’s history and current medical conditions are used to adjust the estimation. Figure 1 shows a very simplified and abstract version of the generator. Technical details are missing and thus the analysis cannot be neither evaluated nor replicated.

6. Given the data that is available to the authors, I would recommend the use of machine learning techniques to estimate the future trajectory and the demand of medication at the personalized level. These methodologies do not require probabilistic assumptions and are able to devise complex, non-linear rules that result in accurate demand estimations.

7. How are the authors evaluating the accuracy of the forecasting methodology they propose? A retrospective analysis of this approach using data from the healthcare system in Italy would provide significant evidence regarding the validity of the proposed generator.

8. To account for the uncertainty in the demand estimation, I suggest to the authors to consider uncertainty sets and optimize against the worst-case scenario. By using robust optimization, they could yield better solutions that protect the hospital system against forecasting errors. For a review of robust optimization approaches, they can resort to the work of Bertsimas et al.:

Bertsimas, D., Brown, D.B. and Caramanis, C., 2011. Theory and applications of robust optimization. SIAM review, 53(3), pp.464-501.

9. The authors should provide a Github repository where researchers can review and access the code and the data from the paper.

Minor Comments

1. The introduction is barely citing other papers that support the authors’ claims. My suggestion would be to add references to other investigations that highlight some of the information that is provided by the authors regarding the drug replenishment at the ICU setting.

2. The first paragraph of the introduction can be split into two paragraphs that capture the two different main ideas discussed in this section.

3. It would be very helpful to the reader to provide an Algorithm overview or a visualization of the entire approach as described in the Introduction.

4. The authors could consider breaking some of the longer sentences in two to help the reader understand better the content of the paper (see Examples in the Introduction and Conclusions).

5. Line 6: Typo – makes it possible continuous review policies.

Reviewer #2: I think this paper is useful and well written.

Moreover, the authors followed rigorous scientific methods.

Here are my main five comments:

1- The paper is missing a proper validation section. This is important.

2- Fig. 1 is not clear, e.g., the path from “Clinician expertise” to “Other Lab Issue” is a Possible death! …

So, I suggest that the flow chart have two exit nodes. One exit is due to death and the other exit is due to (proper) treatment.

3- Hierarchical optimization is a classical problem in Operations Research.

E.g., this is an old classical reference:

Anandalingam, G., & Friesz, T. L. (1992). Hierarchical optimization: An introduction. Annals of Operations Research, 34(1), 1-11.

So, in general, there is no adequate literature review (in the paper) about hierarchical optimization. The aim of this literature review should be to identify the state-of-the-art hierarchical optimization methods to implement one of them (in the paper). Note the implementation of hierarchical optimization in the MILP model is not the state of the art. This is a weak point in the paper.

4- In section 5.3, the authors state the following:

"The number of entities to create is determined by sampling from suitable empirical distributions based on the data collected on-site during the observation period."

I do not understand the term: "suitable empirical distributions".

Moreover, I think it is better first to fit the empirical distribution to a "suitable" theoretical distribution; then to use this theoretical distribution in the simulation model.

5- Also in section 5.3, the authors state the following:

"The algorithm is executed in shell (i.e., while the model runs, the simulation clock does not advance),"

I think the authors mean: (i.e., while the algorithm -- not the model -- runs, ...).

6. PLOS authors have the option to publish the peer review history of their article (what does this mean?). If published, this will include your full peer review and any attached files.

Reviewer #1: No

Reviewer #2: **Yes: **Mohamed Saleh

---

## [Author Response · Author response to Decision Letter 0]

27 Oct 2021

Please find the rebuttal to the Reviewers and the Editors in the attached documents

---

## [Decision Letter · Decision Letter 1]

5 Jan 2022

PONE-D-21-11825R1Rush order containment of critical drugs in ICUsPLOS ONE

Dear Dr. Cappanera,

Thank you for submitting your manuscript to PLOS ONE. After careful consideration, we feel that it has merit but does not fully meet PLOS ONE’s publication criteria as it currently stands. Therefore, we invite you to submit a revised version of the manuscript that addresses the points raised during the review process.

ACADEMIC EDITOR:I would like to thank the authors for the improvements and the revised version. As you will see bellow there are some final points by the reviewer that you need to address before re-submission.

We look forward to receiving your revised manuscript.

Kind regards,

Behzad Behdani

Academic Editor

PLOS ONE

Journal Requirements:

Reviewers' comments:

Reviewer's Responses to Questions

**Comments to the Author**

1. If the authors have adequately addressed your comments raised in a previous round of review and you feel that this manuscript is now acceptable for publication, you may indicate that here to bypass the “Comments to the Author” section, enter your conflict of interest statement in the “Confidential to Editor” section, and submit your "Accept" recommendation.

Reviewer #1: (No Response)

2. Is the manuscript technically sound, and do the data support the conclusions?

Reviewer #1: Partly

3. Has the statistical analysis been performed appropriately and rigorously? 

Reviewer #1: Yes

4. Have the authors made all data underlying the findings in their manuscript fully available?

Reviewer #1: Yes

5. Is the manuscript presented in an intelligible fashion and written in standard English?

Reviewer #1: Yes

6. Review Comments to the Author

Reviewer #1: I would like to thank the authors for their time and effort in conducting a major revision of the manuscript. The paper has been significantly improved as it now provides an indirect comparison of the model’s proposed solution with the actual decisions implemented in practice in June 2016. Section 7 also illustrates an online version of the tool and discusses the implications and challenges associated with it in a real-world setting. The section regarding the demand generation process and forecasting has been expanded and the methodology proposed is now clearer to the reader. Below you will find my comments to the authors. I believe that with a minor revision the manuscript would be ready for publication.

1. In Section 4.1 the authors outline in detail the data sources that they used for their study. Note that Prosafe data were recorded between 01/2011-04/2016 and miss information during the stay. Microbiology data refer to a 6 year period from 01/2010-02/2016 and not a 5 year period as indicated in the paper. The authors also used 69 digitized admissions from June 2019. How was this information subsequently consolidated? Based on the information currently available in the paper, the authors could use the information for patients present in both the Prosafe data and the Microbiology data. However, the rest could not be combined. In addition, the 69 digitized admissions do not refer to the same period (2010-2016) and thus no information from Prosafe and microbiology could be applied for those patients. What was the resulting size of the dataset? How did the authors deal with missing values? It would be great to provide a Table summarizing the mean, standard deviation, and missing percentage for the final dataset. Perhaps this information could go into the appendix, but it remains critically important for the reproducibility of the study.

2. I would suggest the authors to present the R^2 of their computational experiments in predicting length of stay as it provides a comparison with a simple baseline. Perhaps they could consider the average length of stay for each class as the baseline and then analyse the improvement they achieve using the SVR kernel regression.

3. Please add a citation for the Relieff features ranking methodology.

4. The mean of sensitivity and specificity might be misleading for the interpretation of the results. The authors can report instead in Table 6 in different columns the average sensitivity, specificity, and accuracy for a given threshold as well as the overall out-of-sample AUC. The currently presented results are not very insightful from a statistical perspective. In addition, confidence intervals are needed around this metrics.

5. I think it would greatly improve the paper if the authors could add a graphical illustration of the steps necessary to apply the tool. In other words, the timing and sequence of using the forecasting tool and the simulation in combination with the optimization. Parts of that are already described in Section 4.3. This would be a more high-level implementation and workflow of the system compared to Figure 3.

6. What are the implications in terms of cost other than just frequency from the comparisons conducted in Section 6.2? By having better planned and less rush orders what would be the approximate impact on the costing of the department?

7. PLOS authors have the option to publish the peer review history of their article (what does this mean?). If published, this will include your full peer review and any attached files.

Reviewer #1: No

---

## [Author Response · Author response to Decision Letter 1]

18 Feb 2022

Dear Editor,

we addressed all of the Reviewers' comments. A detailed respose to Reviewers is attached.

Sincerely

---

## [Editor Report · Decision Letter 2]

21 Feb 2022

Rush order containment of critical drugs in ICUs

PONE-D-21-11825R2

Dear Dr. Cappanera,

We’re pleased to inform you that your manuscript has been judged scientifically suitable for publication and will be formally accepted for publication once it meets all outstanding technical requirements.

Kind regards,

Behzad Behdani

Academic Editor

PLOS ONE

---

## [Editor Report · Acceptance letter]

28 Feb 2022

PONE-D-21-11825R2 

Rush order containment of critical drugs in ICUs 

Dear Dr. Cappanera:

I'm pleased to inform you that your manuscript has been deemed suitable for publication in PLOS ONE. Congratulations! Your manuscript is now with our production department. 

Kind regards, 

on behalf of

Prof. Behzad Behdani 

Academic Editor

PLOS ONE